# Understanding the development of systematic errors in the Asian Summer Monsoon

Gill M. Martin[1], Richard C. Levine[1], José M. Rodriguez[1] and Michael Vellinga[1]
[1]Met Office, Exeter, Devon, EX1 3PB, UK

*Correspondence to*: Gill M Martin (gill.martin@metoffice.gov.uk)

**Abstract.** Despite the importance of monsoon rainfall to over half of the world's population, many of the current generation of climate models struggle to capture some of the major features of the various monsoon systems. Studies of the development of errors in several tropical regions have shown that they start to develop very quickly, within the first few days of a model simulation, and can then persist to climate timescales. Understanding the sources of such errors requires the

combination of various modelling techniques and sensitivity experiments of varying complexity. Here, we demonstrate how such analysis can shed light on the way in which monsoon errors develop, their local and remote drivers and feedbacks. We make use of the seamless modelling approach adopted by the Met Office, whereby different applications of the Met Office Unified Model (MetUM) use essentially the same model configuration (dynamical core and physical parametrisations) across a range of spatial and temporal scales. Using the Asian Summer Monsoon (ASM) as an example, we show that error

patterns in circulation and rainfall over the ASM region in the MetUM are similar between multi-decadal climate simulations and seasonal hindcasts initialised in spring. Analysis of the development of these errors on both short-range and seasonal timescales following model initialisation suggests that both the Maritime Continent and the oceans around the Philippines play a role in the development of East Asia summer monsoon errors, with the Indian summer monsoon region providing an additional contribution, while the errors over the Indian summer monsoon region itself appear to arise locally. Regional

modelling with various lateral boundary locations helps to separate local and remote contributions to the errors, while regional relaxation experiments shed light on the influence of errors developing within particular areas on the region as a whole.

**Copyright statement**

## 1 Introduction

Despite many advances in weather and climate modelling over the past decades, systematic errors remain prevalent in key regions such as the Asian Summer Monsoon (Sperber et al., 2013). Such systematic errors have been shown in past studies to develop rapidly, often within the first few days of simulation, and can persist to climate timescales (e.g. Martin et al., 2010; Rodriguez and Milton, 2019). This has important implications for forecasting on a wide range of timescales, and for climate projections, in regions where millions of people rely on the seasonal rainfall for their water resources and livelihoods. Several modelling studies have investigated the initial error growth using short-range forecasts from numerical weather prediction models (e.g. Keane et al., 2019; Martin et al., 2010; Rodwell and Palmer, 2007; Phillips et al, 2004). Such studies allow the immediate influence of atmosphere model physical parametrisations to be identified without the complex feedbacks from circulation errors which develop over longer timescales. This approach can be particularly useful where similar model configurations are used for both timescales (Martin et al., 2010; Hurrell et al., 2009).

The advent of coupled ocean-atmosphere numerical weather prediction offers further challenges in the development of additional systematic errors through feedbacks between atmosphere and ocean. For extended-range and seasonal predictions, tracking the development of systematic errors through the coupled atmosphere-ocean-cryosphere system and across timescales ranging from individual weather events through intra-seasonal to seasonal variations is also a challenge. Several previous studies have used initialised seasonal hindcasts to shed light on the origin of coupled model errors in tropical regions (e.g. Lazar et al, (2005); Huang et al. (2007); Liu et al. (2012); Vannière et al. (2014); Siongco et al. (2020)). Lazar et al. (2005) demonstrated that both the atmosphere and ocean components of coupled models contribute to the development of errors, on different timescales and in different regions, and with the balance of atmosphere/ocean contribution being model-dependent. Vannière et al. (2013) used a multi-model seasonal hindcast dataset to identify the order in which errors appeared in the tropical Pacific, and Vannière et al. (2014) developed this into a systematic approach that allowed them to identify a range of drivers and timescales for tropical Pacific SST errors in the IPSLCM5A-LR coupled model. Similarly, Siongco et al. (2020) identified different drivers for the fast-developing cold phase and slow-developing warm phase of the equatorial Pacific SST errors in the Community Earth System Model, version 1 (CESM1). Voldoire et al. (2019) used a multi-model ensemble of seasonal hindcasts made by climate models to confirm that easterly wind stress errors drive warm SST errors in the tropical Atlantic from the first month onwards. In a global study analysing daily to multi-annual timescales in two different coupled seasonal prediction models, Hermanson et al. (2018) showed a range of SST drift evolution and timescales among different regions and different times of year, with some regions being affected by poor initialization.

On sub-seasonal to seasonal timescales, there will be contributions to systematic errors both from local processes and from remote teleconnections. Separating these contributions, and identifying their interaction, requires a range of bespoke modelling tools that constrain parts of the climate system while allowing others to develop freely. Examples include: atmosphere-only, land-only or ocean-only model simulations where observed or modelled fields can be used to force one coupled model component at a time; replacing surface fluxes in a coupled model with daily observed or modelled fields;

regional climate modelling with a range of lateral boundary locations (Levine and Martin, 2018; Karmacharya et al., 2015); global or regional relaxation experiments (Klinker et al., 1990; Rodriguez et al., 2017, 2019); and "pacemaker" experiments

(where a climate model is forced by observed sea surface temperature variations in a specific region, but allowed to evolve freely everywhere else; e.g., Deser et al. 2017; Zhang et al., 2018; Amaya et al., 2019).

Rodriguez and Milton (2019) describe analysis of the spin-up of the errors over the Asian monsoon region in initialized 15-day atmosphere-only hindcasts. These showed the gradual emergence, over the 15 days, of the key systematic errors seen in the moisture transport/divergence from free-running simulations of the same model. Some of the errors are large even at day

1, supporting previous results from e.g. Keane et al (2019) that errors in local parameterised physics are the key drivers of monsoon errors rather than remote forcing errors of the circulation. Rodriguez and Milton (2019) further investigated which errors were driven from the Maritime Continent (MC) region by using regional relaxation experiments where the winds and temperatures over the MC region were relaxed back to reanalyses. This revealed that deficiencies in tropical convection over the MC region start to contribute to errors in the Asian monsoon circulation within the first 15 days of the hindcasts. Levine

and Martin (2018) used a regional climate model centred over India and forced by reanalyses at the lateral boundaries to show that remote errors (particularly, excessive convection over the equatorial Indian Ocean and poor representation of precursor disturbances transmitted from the Western Pacific) contribute significantly to the poor simulation of monsoon low pressure systems in the Met Office model.

In the present study, we illustrate how a combination of many of the techniques outlined above can be used to analyse the

development of monsoon errors, their local and remote drivers and feedbacks. We take advantage of the range of Met Office model configurations covering timescales from days through seasons to decades. These share a common dynamical core and similar physical parametrisations as part of the Met Office's seamless approach to modelling weather and climate. We extend and develop the previous work by including analysis of the development of errors in medium-range coupled and atmosphere-only model hindcasts during the first 7-15 days, and in a coupled seasonal hindcast ensemble during the first few

pentads following initialization, and by investigating the individual and interacting roles of various remote regions in the development of errors in both atmosphere-only and coupled model configurations. While our study focusses on systematic errors in the Asian summer monsoon, similar methods could be applied to other monsoon, and non-monsoon, regions. Section 2 describes the data and methods used, while Sect. 3 documents the results of the various experiments. In Sect. 4 we discuss the results and their implications for targeted model development.

**2 Data and methods**

The model configurations and simulations used in this study are summarised in Table 1. Free-running climate simulations using Met Office coupled atmosphere-ocean configuration Global Coupled version 2 (GC2.0; Williams et al, 2015), forced by present-day greenhouse gases and aerosols and covering several decades, are used initially in order to illustrate the model errors of interest to this study. The atmosphere component of GC2.0 is Met Office Unified Model (MetUM) Global

Atmosphere 6.0 (GA6.0; Walters et al., 2017), which is coupled to the Joint UK Land Environment Simulator (JULES; Best et al., 2011) and the NEMO (Nucleus for European Modelling of the Ocean; Madec, 2008) ocean and CICE (Hunke and Lipscomb, 2004) sea-ice models. The model is configured at resolution 0.833° longitude x 0.556° latitude (which is approximately 80 km at the equator) in the horizontal for the atmosphere and the ORCA0.25 tripolar grid for the ocean. On points of regular zonal and meridional grid spacing the ocean resolution is 0.25°, while the tripolar grid ceases to be a regular

grid on points poleward of 20°. The vertical resolution is 85 levels for the atmosphere and 75 levels for the ocean. Comparison is made against ERA-interim (ERA-I; Dee et al., 2011) and ERA5 (Copernicus Climate Change Service, 2017) reanalyses for winds, the Global Precipitation Climatology Project pentad dataset version 2.2 (GPCP v2.2; Xie et al., 2003; Adler et al., 2003) and the Tropical Rainfall Measuring Mission 3B42 product, version 7-7A (TRMM; Kummerow et al., 1998; Huffman et al., 2010; Huffman and Bolvin, 2013) for precipitation, and NOAA Optimum Interpolation Sea Surface

Temperature v2 (OISSTv2; Reynolds et al. 2007).

In order to study the development of errors after initialisation, we make use of a hindcast ensemble from the GloSea5 operational long-range forecast system (MacLachlan et al., 2015; Williams et al., 2015). GloSea5 also uses the MetUM GC2.0 configuration at the same horizontal and vertical resolution as in the free-running simulations described above. The standard operational hindcast set includes seven members per start date, four start dates (1, 9, 17, 25) per month, and runs

from 1993-2016. A Stochastic Kinetic Energy Backscatter scheme (SKEB2; Bowler et al. 2009) is used to introduce small grid-level perturbations throughout the integrations to create ensemble spread. The atmosphere and land components are initialized from daily ERA-I reanalyses at 0.75° ×0.75° resolution, while the ocean and sea ice models are initialized from the GloSea5 ocean and sea ice analysis using GloSea5 Global Ocean 3.0, which is driven by ERA-I and uses the NEMOVAR data assimilation scheme (Blockley et al. 2014).

In order to separate the influence of local and remote sources of error, we make use of a regional climate model (RCM) configuration, forced at the lateral boundaries by 6-hourly ERA interim re-analyses and at the surface by observed sea surface temperatures (SSTs) from OISSTv2, and run at 0.44° × 0.44° resolution (approximately 50 km, which is similar to (but slightly higher than) that of the GC2.0 and GloSea5 global models). The RCM was only configured at GA7.0 (Walters et al., 2019), which includes changes, from GA6.0, to both model physics and dynamics that are both incremental

developments and targeted improvements to address critical errors that included a persistent dry bias over the Indian subcontinent. While some progress was made in GA7.0 global models towards reducing those errors, the overall pattern of ASM errors investigated in the present study remains, justifying our use of this RCM configuration. For the RCM domains centred over China, a rotated north pole is used at 61°N, 296.3°E. The RCM is constrained by 6-hourly ERA-I at the lateral boundaries, but within the domain the model runs freely after initialisation and is therefore able to develop errors due to local

processes and feedbacks, despite the constraint from the boundaries. By adjusting the locations of the lateral boundaries and comparing between RCM simulations, and against a corresponding 20-year atmosphere-only global climate model (AGCM) GA7.0 simulation at resolution 0.833° longitude x 0.556° latitude (denoted AGCM-N216), the contribution to the systematic errors from different regions can be ascertained. This technique was applied by Karmacharya et al. (2015) and Levine and

Martin (2018) for understanding sources of error in the mean state, intraseasonal variability and monsoon low pressure
systems in the South Asian Summer Monsoon (SASM). Hence in this study we centre our RCM domain over China to investigate the sources of error in the East Asia Summer monsoon (EASM).

We study the evolution of errors after initialisation in the GloSea5 season hindcast ensembles with different start dates and, in addition, initialised 7 to 15-day numerical weather prediction (NWP) hindcasts using atmosphere-only and coupled configurations of GA6.1[1]/GC2.0 at resolution 0.234° longitude x 0.156° latitude (approximately 26 km at the equator). These
are initialised every day between 1st May and 19 September 2016 and each run for 15 days (Vellinga et al., 2020). The day-1, day-2, etc hindcasts can be combined to provide a seasonal climatology for each lead time, and the use of the same atmosphere model configuration in the coupled and atmosphere-only hindcasts allows the role of coupling to be ascertained.

To shed light on the drivers of systematic errors, we make use of the "nudging" technique described by Rodriguez and Milton (2019). A 20-year, free-running, atmosphere-only, model simulation using GA7.0 at resolution 1.88° longitude x
1.25° latitude (approximately 200 km at the equator, denoted AGCM-N96) is relaxed back to analyses over regions from where we consider significant systematic errors may originate and affect other regions through remote teleconnection. Model winds and potential temperatures are nudged back to ERA-I with a 6-hourly relaxation time scale at all model levels. A 10° buffer zone around the relaxation subdomain is applied in which the nudging increments are exponentially damped to zero, in order to ensure a smooth transition between the nudged and free-running parts of the simulation. Similar nudging
experiments are also carried out in 15-day hindcasts initialised once per day through JJA of 2016 using the NWP GA6.1 atmosphere-only configuration at resolution 0.833° longitude x 0.556° latitude (denoted NWP-2016; see Table 1), in order to examine how the influence from the nudged region is manifest in the development of the errors.

## 3 Results and discussion

### 3.1 Climatological errors in the Asian summer monsoon

Figure 1a shows June to August (JJA) mean climatological errors in rainfall and 850 hPa winds in the 30-year, free-running, present-day, GC2 simulation compared with ERA-I and GPCPv2.2. Similar to previous studies of the Asian monsoon system in MetUM configurations (e.g. Keane et al., 2019; Johnson et al., 2016, 2017; Bush et al., 2014; Martin et al., 2010; Ringer et al., 2006) the model exhibits a deficit in rainfall over the Indian peninsula, the eastern Indian Ocean south of the Equator and the Maritime Continent, an excess over the Indian Ocean to the north of the Equator, in the eastern South China Sea
(SCS) and the western Pacific, and excess precipitation over the mountains bordering the Tibetan plateau. These are accompanied by a weak Somali jet that diverges into an anticyclonic anomaly over India, excessive westerly flow over

---

[1] This operational NWP configuration includes a small number of scientific differences from GA6.0; see Walters et al. (2017).

southeast Asia, the SCS and across the Philippines into the western Pacific, and a cyclonic error and deficit in precipitation over southeastern China, southern Japan, Korea and the East China Sea.

Johnson et al. (2017) analysed the climatological June to August (JJA) seasonal mean errors in a hindcast ensemble from GloSea5 and showed that they are similar to those seen both in climate models including the MetUM (Sperber et al., 2013) and in other state-of-the-art seasonal forecast systems. Figure 1b shows the JJA climatological errors from the current GloSea5 23-year operational hindcast ensemble initialised each year on the four start dates in April. The pattern correlation for rainfall errors between Fig.s 1b and 1a is 0.88. This confirms once again that, despite the initialisation and the relatively short lead time, the hindcast JJA errors are very similar in pattern, and (with the exception of the Indian region) in magnitude, to those from the 30-year free-running simulation (Fig 1a).

Johnson et al. (2017) commented that the seasonal mean errors over the Indian region are largely due to a climatologically late onset of the monsoon in the model, which reduces the precipitation over and around India in May and June. Figure 1(c to h) shows errors in the June, July and August climatologies from GC2 and from the GloSea5 23-year operational hindcast ensemble (at ~1 month lead time for each month, i.e. using the 4 start dates in May, June and July respectively; note that this lead time is shorter than for the JJA mean shown in Figure 1b). This shows that, while the rainfall errors over the Indian region as a whole in both GC2 and GloSea5 are indeed largest in June, and the pattern correlations between the rainfall errors in GC2 and GloSea5 for June (0.66), July (0.81) and August (0.87) are high, both the magnitude of the monthly errors and the differences between the three months are noticeably smaller in GloSea5.

The differences between GloSea5 and GC2 in the Indian region particularly in June (GC2 shows a weakened Somali Jet and much larger rainfall deficit than GloSea5) are consistent with the differences between GloSea5 and CMIP5 models commented by Johnson et al. (2017), who considered these attributable in part to a smaller northern Arabian Sea cold SST error in GloSea5. Figure 2 shows the errors in SST against OISSTv2 observations (1993 to 2015) in JJA and for June, July and August for GC2 and GloSea5 as in Fig. 1. Cold errors in the northern Arabian Sea are seen in both simulations, particularly in June, but they are considerably larger in GC2. Marathayil et al. (2013) showed that, in CMIP3 models, such errors develop in winter due to anomalously strong north-easterly winter monsoon winds advecting cold, dry air from the Eurasian land mass over the Arabian Sea. Their analysis suggested that excessive rainfall in the equatorial Indian Ocean and anomalously cold winter continental surface temperatures in the CMIP3 models both contribute to the northern Arabian Sea cold SST error. Levine et al. (2013) showed that these errors persist into spring and early summer and are associated with a weaker monsoon circulation and reduced monsoon precipitation. Initialisation of the GloSea5 hindcasts in spring prevents the growth of a large SST error, thereby reducing the circulation and rainfall errors over the Indian region (Levine and Turner 2012; Levine et al. 2013, personal communication R. Levine).

The free-running and initialised models are consistent in developing cold SST errors around the Maritime Continent, the South China Sea and the central and eastern Indian Ocean, and warm SST errors in the western Indian Ocean, even just over a month after initialisation. An SST error dipole pattern resembling that of the positive Indian Ocean Dipole (IOD; Saji et al., 1999) is apparent in the seasonal hindcasts but is much stronger in the free-running simulation. This is consistent with the

circulation anomaly pattern shown in Fig. 1 which strongly resembles the atmospheric component of the IOD teleconnection: south-easterly anomalies along the Sumatran coast and easterly anomalies along the Equator. Previous work (e.g. Marathayil et al., 2013; Johnson et al., 2017) has shown that this SST error pattern is associated with a coupled interaction between excessive rainfall in the central equatorial Indian Ocean, excessive easterly low-level winds and increased upwelling that shoals the thermocline in the east. The additional northeasterly wind anomalies in the western Indian Ocean in GC2 exacerbate this error pattern. Johnson et al. (2017) showed that this coupled mean state error results in errors in the representation of the IOD as a mode of variability in the model, reducing its ability to predict the Indian monsoon circulation.

Rodriguez and Milton (2019) showed that local errors in moisture convergence/divergence over the Maritime Continent region also contribute to the development of circulation and rainfall errors in the eastern Indian Ocean, the South China Sea, western Pacific and southeast China in atmosphere-only simulations. It is likely that, in the coupled system, these atmosphere errors drive a cooling response in the SSTs which further contributes to decreases in rainfall and anomalous moisture divergence through coupled feedbacks.

The largest differences between the free-running and initialised simulations are seen in the central North Pacific, where the cold SST errors in the free-running GC2 simulation are much larger than in GloSea5. Such errors are common among CMIP5 models: Wang et al. (2018) showed that they are associated with overly strong surface winds driving excessive evaporation, combined (in summer) with a deficit in downward solar radiation at the surface. While Wang et al. (2018) showed that, in CMIP5 models the cold errors are present throughout the year (but largest during JJA), the initialisation of GloSea5 in spring limits the extent to which these can develop during the summer months. Overall, the pattern correlations between the SST errors in GC2 and GloSea5 for the region shown, in June (0.52), July (0.59), August (0.68) and JJA (0.66), are moderate.

Despite the differences related to errors which develop in the winter in GC2, as discussed above, there are many areas where the similarity between the monthly error patterns at ~1-month lead time and the seasonal mean error pattern demonstrates that the errors develop quickly and then persist to longer timescales in this coupled model. In the following sub-sections we demonstrate how a range of configurations within the seamless modelling system can be used to shed light on various aspects and drivers of these errors.

### 3.2 Regional climate modelling

To investigate first the local and remote sources of some of the errors identified in Sect. 3.1, we use regional climate model (RCM) simulations with different domain sizes, centred over China and forced at the lateral boundaries with ERA-I 6-hourly re-analyses, and using time-varying, observed SSTs instead of an ocean model. Such experiments isolate the effects of any remote errors in an AGCM that are located outside the RCM domain from those developing within the domain (with the goal of testing the effects of the simulated climate in different remote areas on the central area of interest). The RCM simulations are performed at 0.44° x 0.44° resolution, similar to that of the AGCM-N216 simulation (0.833° × 0.556°), so that the

comparison between RCM and AGCM isolates the local and remote forcing of errors over these two regions during the
225 EASM. Karmacharya et al. (2015) used this approach to investigate local and remote sources of MetUM errors in the SASM
region. They showed that the equatorial Indian Ocean is a key driver of Indian rainfall errors, although errors over the
Himalayan foothills also played a role and there was evidence of locally-driven errors that were thought to be related to the
model's inherent difficulties in reproducing the diurnal cycle of rainfall over land. Levine and Martin (2018) used similar
methods to show that remote errors contribute significantly to the poor simulation of Indian monsoon lows and depressions.

The RCM domains used in the present study are shown in Fig. 3. While some domain boundaries cross through the steep
orography of the Himalayas, producing some erroneous values in close proximity to the boundary, such effects occur far
enough away from the area of interest (China, in this case) to have minimal influence on the results. Figure 4 shows the
climatological errors in JJA from the AGCM-N216 and the RCM China1 and China1SE (which includes the China1,
China1E and China1S regions) domains. Although the magnitude of the error differs in places, the error pattern for JJA in
the AGCM-N216 (top left) is very similar to that seen in the coupled simulations (Fig. 1a); the pattern correlation between
the rainfall errors in AGCM-N216 and those in GC2.0 for JJA (over the region shown in Fig 1a) is 0.70. This suggests that
neither the change between GA6.0 (the atmosphere component of GC2.0) and GA7.0 (used in AGCM-N216), nor the
atmosphere-ocean coupling, has a major impact on the overall error pattern. This is consistent with Walters et al. (2019) and
Williams et al. (2017) who showed reductions in the magnitude of in the JJA rainfall bias in this region between GA6.0 and
240 GA7.0 (attributed to the inclusion of a stochastic physics package and changes to convection and warm rain microphysics)
and between GC2.0 and GC3.0, respectively, but little change in their pattern.

The local RCM simulation, China1 (Fig. 4, upper right panel), favours southerly and south-westerly anomalies over
southeast China, the South China Sea and the Philippine Sea, and a more widespread wet error than the AGCM. This
includes much of the steep orography along the southern and south-eastern edges of the Tibetan Plateau, and most of
245 southern China. The circulation anomalies over southeast China and the surrounding seas in China1 are opposite to those in
the AGCM, suggesting that the characteristic error of a weakened southwesterly flow and rainfall deficit over southeast
China and the East China Sea are not locally driven.

Extending the domain to the south and east (China1SE) contributes north-easterly anomalies and a dry error over the
middle/lower Yangtze River Basin (Fig. 4 lower right panel), although this does not recover the full AGCM error (Fig. 4,
lower left panel). Neither of the experiments using the China1E and China1S domains individually contribute northeasterly
anomalies (Fig. 5 lower middle and centre right panels), indicating that the source of these anomalies is the south-eastern
sector of this domain. However, the eastward extension in China1E does produce an anomalous low-level easterly wind
component towards eastern China as part of an anomalous cyclonic flow over the western Pacific, representing a weakening
and northward displacement of the Western North Pacific Subtropical High (WNPSH). There is associated drying over land
and increased rainfall offshore. The southward extension in China1S contributes weakly both to the westerly anomalies
across the South China Sea and to the easterly anomalies over the middle/lower Yangtze River Basin, but both of these are
strengthened when the domain is extended both to the south and the east, thereby including in addition the whole of northern

Indonesia. Extending the China1 domain to the south and west to include more of the Indian monsoon region, including the Arabian Sea and part of the western equatorial Indian Ocean (China1W and China1SW) tends to promote dry anomalies compared with those contributed locally by China1 itself (Fig. 5), while drying and anticyclonic anomalies emerge over the Indian region (particularly in China1SW) that are similar to the climatological errors in this region seen in Figure 1. Extending the domain northwards (China1N) contributes additional southwesterly anomalies and some drying in southern China. Preliminary experiments with much larger RCM domains (not shown) suggest a role for even more remote influence, perhaps through the circum-global teleconnection (Wu et al., 2016). The main effects of the different domain extensions analysed are summarised in Table 2.

A limited number of RCM experiments was carried out in which the RCM was initialised each year on 1 May, in order to determine how quickly the influence of local processes and remote teleconnections became apparent. For all domains, the differences between the re-initialised and free-running experiments were minimal (not shown), indicating a rapid and robust evolution of the atmosphere model towards these systematic errors. The development of errors in the first few weeks after initialisation is explored further in the next section.

This analysis illustrates how an RCM with different lateral boundary locations can be used to shed light on the local and remote sources of systematic error in a climate model. For the EASM, we find that much of the circulation and rainfall error pattern seen in the full GCM is not driven locally but is related to errors arising mainly to the south and east of the region, i.e. over the Maritime Continent, South China Sea and the Western Pacific. This is in contrast with the previous published studies using this technique over the SASM region (described above) which indicated more local sources to many of the errors. Levine and Martin (2018) showed that the inclusion of East Asia in the domain centred over India made very little difference to the mean state errors over India.

### 3.3 Development of errors in initialised seasonal hindcasts

Having identified that in much of the ASM region the errors appear to develop rapidly and persist thereafter to long timescales, we now demonstrate how initialised hindcasts can be used to examine their development and evolution during the first few weeks after initialisation. We first make use of the GloSea5 seasonal hindcast ensemble, which consists of 7 members per start date for four start dates per month and covers a 23-year period from 1993-2015. In order to reduce the effects of internal variability, we average the ensemble mean precipitation, winds and SSTs into pentads and average both the model and observational fields over the hindcast period.

### 3.3.1 ASM region as a whole

Figures 6 and 7 show the climatological development of rainfall, wind and SSTs errors in the seasonal hindcast ensemble, pentad by pentad, following initialisation on 25th May. Anomalously warm SSTs and excess precipitation occur in the SCS and western Pacific soon after initialisation, with cold anomalies and deficient precipitation around the Maritime Continent. The circulation anomalies initially show divergence over the Maritime Continent and southerly anomalies into the SCS. As

the hindcasts progress, the cold/dry anomalies around the Maritime Continent expand westwards and northwards and the southerly anomalies develop into westerly anomalies that form the southern flank of an anomalous cyclonic pattern over the western Pacific which represents the weakening and displacement of the WNPSH. The westerly anomalies intensify as the SASM onsets at the start of June. This is related to anticyclonic anomalies that develop over India rapidly after initialisation and are associated with a weakening of the SASM trough, combined with excessive rainfall over the steep orography of the eastern Himalaya that promotes drying over the head of the Bay of Bengal. Levine and Martin (2018) showed that the MetUM typically underestimates the number, and rainfall contribution from, monsoon lows and depressions, which also are unable to progress across northern India. In the absence of these features, rainfall over the Bay of Bengal is reduced and that over the Myanmar orography is increased, with an associated acceleration of the westerly flow across the Bay of Bengal and southeast Asia into the South China Sea. This converges with the southerly anomalies from the Maritime Continent region, promoting further rainfall and creating a positive feedback that develops the westerly wind error (extension of the westerly jet) across the SCS and the Philippines into the western Pacific. A twin cyclonic error develops over the SCS and western Pacific in early June, developing northwards and causing the northeasterly anomalies over southeast China that were highlighted in previous sections. Positive rainfall errors also appear over the Indian Ocean to the south of the Indian peninsula in mid/late May, associated with the anomalous northerly winds from the peninsula (and an increasing dry error there) converging with the anomalous easterly winds from the eastern equatorial Indian Ocean (EEIO). By mid-June, the patterns of rainfall and wind errors closely resemble the long-term June climatological errors in the free-running simulations (Fig. 1 and Fig. 4).

Figure 7 shows similar analysis of the SST errors. Warm anomalies develop immediately in the SCS and western Pacific, associated with the positive rainfall errors seen in Fig. 6. Warm anomalies also develop in the Bay of Bengal soon after initialization, particularly in the head of the Bay where the rainfall and cloud are reduced and warm anomalous winds converge from northern India. Cold anomalies develop around the Indonesian islands and over the northern and western edges of the Arabian Sea. Over the first 15 days of the hindcast, the south-eastern Bay of Bengal warms while the cold anomalies around the Indonesian islands spread westwards and northwards, temporarily creating a weak north-south dipole across the Equator in the EEIO. However, in subsequent pentads the warm anomalies are replaced with colder SSTs, in association with increasing south-easterly anomalies along the Sumatran coast and diverging 850 hPa wind anomalies and a negative rainfall error across the whole of the Maritime Continent. Warm SST errors develop in the western equatorial Indian Ocean and central and eastern Arabian Sea by mid-June, creating the east-west dipole error pattern seen in the long-term climatological errors, while cold errors persist along the northern and western Arabian Sea coasts.

We find that the error patterns develop in a similar way when using start dates of 25th June, 25th July and 25th August (not shown). This indicates that they are a robust feature of the model's behaviour during the monsoon season, consistent with their similarity to those in the free-running coupled simulation. We now examine the error evolution in different parts of the ASM region separately.

### 3.3.2 East Asian Summer Monsoon

Figure 8 shows pentad rainfall, winds and SSTs averaged over various different regions, from hindcast ensembles initialised on different start dates between 9th April (0409) and 25th August (0825), along with similar timeseries for GPCP and TRMM rainfall observations, ERA-I winds and OISSTv2 SSTs. For start dates in April, the SST in the SCS initially warms excessively, before cooling systematically into a cold error through the JJA season (Fig. 8a). For start dates in late May onward, the SST appears to be initialised systematically warmer than the observations but to cool thereafter. The peak warm SST error coincides with the broadscale seasonal transition that is heralded by the South China Sea summer monsoon onset, as determined by the reversal of the 850 hPa winds over the SCS that (climatologically) occurs during pentad 28 (see Figure 9) in the criterion suggested by Wang et al. (2004). The SST cooling after this transition coincides with an acceleration of the westerly winds into a positive error for all start dates (Fig. 8b). In response both to this and to the additional convergence of moisture into the SCS from the Maritime Continent, the rainfall over the "Philippines" region (Fig. 8c) starts with a positive error and increases thereafter, particularly in the hindcasts initialised in May and June.

The East Asian monsoon Index (EASMI: see Wang et al., 2008) decreases rapidly after initialisation in all hindcasts (Fig 8d), indicating the weakening and displacement of the WNPSH. Separation of this index into its two components (red boxes on Fig. 9) reveals that this is driven mainly by the increasingly excessive westerly flow in the southernmost box (dot-dash lines in Fig. 8d), which extends from southeast Asia across the SCS and the Philippines into the western Pacific, including the SCS box in Fig. 8b, and largely coinciding with the "Philippines" region in which the rainfall also increases (Fig. 8c). However, Fig. 6 shows that the hindcasts also rapidly develop an easterly error in the northernmost box (dashed lines in Fig. 8d), which extends from southern China across the East China Sea and to the south of Japan. This is a characteristic systematic error of the EASM in Met Office models and is associated with a lack of northward advancement of the Meiyu rain band and a deficit in the seasonal mean rainfall (e.g. Zhang et al., 2020; Martin et al, 2020). This easterly error is the northern part of the cyclonic anomaly that begins in the SCS in response to anomalous divergence from Indonesia and expands northwards and eastwards as the anomalous westerlies spin up over the "Philippines" region. This analysis confirms the suggestion from the RCM simulations in Sect. 3.2, that the errors in the EASM over China are driven largely by errors arising to the south and east of the region. This will be explored further using relaxation experiments in section 3.5.

### 3.3.3 Equatorial Indian Ocean

Figure 7 showed that, in hindcasts initialised from late May onwards, there is an initial slight warming in the Bay of Bengal which is replaced by a slight cooling within the first 6 pentads. Cooling of the SSTs within the Maritime Continent spreads westwards to the south of the Equator while warmer SST anomalies develop in the western Indian Ocean, both in response to increasing westerly wind anomalies from the EEIO. However, similar analysis using start dates in late April and early May shows a slightly different development in the first ~15 days of the hindcast (see Fig.s 10 and 11): the anomalous divergence and rainfall deficit over the Indonesian islands is much more localised and takes longer to spread westwards and northwards.

There is greater and more widespread warming of the southern Bay of Bengal, while the cold anomalies south of the Equator off the Sumatran coast do not start to develop until around 20[th] May. Figure 8e confirms that the SSTs to the north of the Equator (red dashed box on Fig.s 10 and 11, last panel) warm substantially over the first few pentads for all of the April and early May start dates before cooling and ultimately developing a cold error, while for later start dates there is only a short period (2 or 3 pentads) of initial slight warming before a similar cooling begins and persists for the rest of the season. In contrast, the SSTs in the EEIO to the south of the Equator (Fig. 8f; red solid box on Fig.s 10 and 11, last panel) cool systematically for all start dates.

Examination of the 850 hPa winds in Fig. 10 shows that the initial warming in the EEIO box to the north of the Equator for the late April start date is associated with a developing rainfall excess in the western Indian Ocean and deficit in the east. This is followed by westerly anomalies along the equatorial region and, by pentad 27, northeasterly anomalies from southeast Asia, opposing the mean flow (Fig. 9). After pentad 28, this is replaced by westerly/southwesterly anomalies (accelerating the mean flow) and a developing positive rainfall error, with an additional inflow to the region from southeasterly anomalies along the Sumatran coast. For start dates after this seasonal transition (Fig. 6, Fig. 8e) the wind anomalies are persistently westerly/southwesterly with an increasing positive rainfall error and cooling SSTs. This is thought to be once again related to the seasonal transition that takes place around mid-May and marks the start of the Asian monsoon season (Wang et al. (2004); Fig. 9). Prior to this transition, the mean state low-level winds over the equatorial Indian Ocean are westerly and the mean flow over the Indonesian islands is weak. As noted by Ding and Chan (2005), the onset of the South China Sea summer monsoon is very abrupt, with a rapid switch from easterlies to westerlies over the South China Sea and a rapid expansion north-eastwards of the south-westerlies from the EEIO across the Indochina Peninsula. In hindcast ensembles initialised after this transition (Fig. 6, 7), when the easterly low-level flow over the EEIO south of the Equator is stronger, there is a more widespread anomalous divergence over the Maritime Continent and more rapid cooling of the SSTs to the west of Sumatra. This analysis illustrates that the monsoon error development in initialised hindcasts can be dependent on the stage of the monsoon season, as well as on the lead time of the hindcast. Once the broadscale seasonal transition has occurred, the error patterns develop in a similar manner regardless of the initialisation date.

The development of errors in the northern Bay of Bengal and Arabian Sea is also somewhat different in the hindcasts initialised before and after the seasonal transition. In the earlier-initialised hindcasts (Fig. 10), excessive rainfall appears over the Eastern Himalaya soon after initialisation and is associated with anomalous convergence from the south that combines with developing northwesterly anomalies over northern India into an anticyclonic error and deficient rainfall over the whole Bay. This persists until the seasonal transition at pentad 28 and thereafter (as discussed above) develops into westerly anomalies in a similar way to the hindcasts initialised on 25[th] May (Fig. 6). At the same time, anticyclonic errors also develop over the Arabian Sea as the positive rainfall error forms in the western equatorial Indian Ocean and the mean westerly flow over the equatorial Indian Ocean that turns into south-easterly flow off the coast of Somalia (Fig. 9) weakens (Fig. 10). Cooling of SSTs in the northern Arabian Sea is present soon after initialisation and develops into a larger and more widespread cold error than in the later-initialised hindcasts. Marathayil et al. (2013) suggested that such cooling (in winter

and spring) is related to advection of too-dry and too-cold air from Pakistan and surrounding regions, aided by erroneous
strengthening of the winds (seen in Fig. 10), leading to excessive evaporation. However, once again, this pattern changes
after the seasonal transition and ultimately develops in a similar way to the later-initialised hindcasts.

## 3.4 Development of errors in initialised NWP hindcasts

Further information on the development of these errors can be gleaned through the use of hindcasts from the Met Office's
NWP model, in both atmosphere-only and coupled configurations. The atmosphere-only runs (UNCPLDNWP, see Table 1)
are 7-day operational forecasts, while the coupled model hindcasts (CPLDNWP) are run for 15 days. In both cases, there was
one ensemble member per day run in near-real time since 1 May 2016. Results shown in Sect. 3.2 indicated that the errors
developing in atmosphere-only configurations closely resemble those in the coupled atmosphere-ocean models. However,
SST errors are also identified, so it is important to understand the extent to which these are driven by, and feed back upon,
the atmospheric errors. As an example of the use of these NWP hindcasts, we examine the development of SST and wind
errors in the Indian Ocean region, and their sensitivity to horizontal resolution, in a 2016 case study.

### 3.4.1 Influence of horizontal resolution

The atmosphere components of both UNCPLDNWP and CPLDNWP are configured at 0.234° x 0.156° resolution, which is
considerably higher than that in GC2.0 and GloSea5-GC2, so we first consider how this affects the error development. As
discussed above, the development of the errors in parts of the ASM region over the first ~15 days of hindcast differs
according to whether the hindcasts are initialised before or after the broadscale seasonal transition. This is further illustrated
by composite analysis of SST and 10m winds at forecast lead times of 1, 5 and 15 days of the CPLDNWP hindcasts, over a
period of 10 to 15 days on either side the broadscale seasonal transition (Fig. 12). For 2016, the validity dates chosen are 10
to 19 May ("before") and 10 to 23 July ("after").

Figure 12(a-c) shows the emergence of SST and surface wind errors in the Indian Ocean before the transition. At day 1 the
biases are small, showing in part the discrepancies between the OISSTv2 SSTs and the analysis SSTs (FOAM, Waters et al.,
2014) used to initialise the hindcasts. At longer lead times, a large warm bias develops in the EEIO and the SCS, which is
associated with a weakening of the equatorial westerly flow and the southerly wind in the Bay of Bengal. At the same time,
cold SSTs develop in the northern Arabian Sea and southwesterly wind anomalies develop along the Somalian coast, in a
similar manner to that seen in GloSea5 (Fig. 10, 11). The emergence of errors after the transition (Fig. 12d-f) shows a
different pattern. A cool bias starts to develop in the Maritime Continent and the adjacent ocean in the southern Bay of
Bengal. The error is associated with a zonal wind anomaly extending from the Bay of Bengal to the tropical Western Pacific
and southeasterly anomalies along the Sumatran coast and in the central equatorial Indian Ocean. On the other hand, a warm
bias develops in the equatorial western Indian Ocean, connected with a weakening of the surface wind in the region. These
results are consistent with the analysis of GloSea5 (Fig.s 6, 7, 10, 11), despite the greater atmospheric horizontal resolution
used in the CPLDNWP hindcasts which has the potential to reduce the errors through the improved representation of

orography and coastlines. This confirms that the error patterns emerging in the first 15 days, both before and after the broadscale transition, are robust and largely insensitive to horizontal resolution.

### 3.4.2 Evolution of SST errors in the EEIO in coupled and uncoupled hindcasts

The change in evolution of the SST errors in the northern EEIO box (as used in Fig. 8e) over the first 15 days of the
CPLDNWP hindcasts initialised between May and early August (Fig. 13a) is also similar to that seen in GloSea5. In forecasts initialised in May, SST in the northern EEIO box develops a warm error of around 0.5°C relative to both the ocean analyses used to initialise the coupled forecasts (FOAM, Waters et al., 2014) and to OISSTv2 observed SSTs. This warm error manifests itself as a tendency to under-predict the cooling of SSTs in the second half of May. Forecasts initialised in June and July do not have this problem and develop a much weaker warm SST error within the first 15 days, mostly around
0.1°C, again consistent with the results for GloSea5. In fact, SSTs follow the observed cooling and levelling-off during June and July reasonably well. The warming of SSTs relative to the ocean analyses during the second half of May stems, at least partly, from under-representing the cooling that is seen in ERA5. That cooling is related to increased surface heat loss during that period (Fig. 13d). However, in CPLDNWP, excessive downward solar radiation (not shown) and under-estimated turbulent (i.e. latent and sensible) heat fluxes (e.g. Fig. 13e) contribute to a reduction in the net surface flux out of the ocean
during this period. The error in turbulent fluxes can be partly traced to a weak surface wind error (Fig. 13c). Errors in ocean processes likely also contribute to SST errors. These may be surface-driven (due to the weak surface wind bias, Fig. 13c) or caused by deficiencies in ocean processes (e.g. vertical mixing). Shallow errors in ocean mixed layer depth would exacerbate warming of SST caused by surface flux errors. Figure 13b confirms a lack of deepening of the mixed layer in the model early in the period, consistent with the weak-wind bias during that period. In future work we will examine the contribution
from ocean processes in more detail.

By comparing surface heat flux errors from coupled and atmosphere-only forecasts in this period we can determine the importance of air-sea coupling in the development of surface flux errors (Fig. 13d,e). For most of the time, the evolution of surface flux errors is very similar between coupled and uncoupled configurations. This suggests that coupled feedbacks are of limited importance here in the development of surface flux errors. The main exception is during the second half of May,
when the strongest warm SST error develops. In this period, the differences in net surface heat flux and surface latent heat flux error between CPLDNWP and UNCPLDNWP are unusually large, differing by 50-100 W m$^{-2}$ (Fig. 13e). Coupled feedbacks cause reduced latent heat loss in CPLDNWP, compared to ERA5 (positive values in Fig. 13e), while UNCPLDNWP shows excessive cooling from surface latent heat flux (negative values in Fig. 13e), consistent with a positive 10m wind bias in UNCPLDNWP (Fig. 13c). Further work is needed to clarify how this coupled feedback operates,
including the use of targeted sensitivity tests in order to separate the different components. This example illustrates how coupled and uncoupled initialised forecasts can be used to home in on some of the long-standing errors seen the Indian Ocean.

### 3.5 Regional nudging experiments to assess sources of error

From the analysis shown in previous sections, we hypothesise that the reduced rainfall and anomalous outflow from the Maritime Continent and Indian regions play a role in the development of the circulation errors in both the EASM and the Indian Ocean at the start of the monsoon season, while the errors in the SASM region appear to arise locally. In order to test this hypothesis, we conduct a series of atmosphere-only sensitivity experiments using the nudging/relaxation methodology described in Rodriguez et al. (2017). This involves relaxing the temperatures and winds back to analyses with a 6-hourly relaxation time scale at all model levels. Assuming a linear response, the difference between the Control and the "Nudged" simulations then gives an indication of the role played by the nudged region in the errors that occur in the Control in other locations (Klinker 1990).

### 3.5.1 Free-running simulations

We apply this methodology first to climate simulations, using the GA7.0 atmosphere-only configuration AGCM-N96 (see Table 1). We use four different nudging regions, referred to as the "Philippines", "Indonesia", "South Asian Summer Monsoon" (SASM) and "Maritime Continent" (MC) regions. These regions were chosen based on the analysis in sections 3.2 and 3.3 which indicated that the Maritime Continent region may be influencing the development of errors in the EASM, and that the Philippines and Indonesia region may contribute both independently and jointly. For the SASM region, previous published studies using the RCM had indicated that many of the errors were locally-driven and had only a minor influence on the wider ASM, so the influence of nudging in this region is also examined. For these experiments, the horizontal winds and potential temperature at all model levels are relaxed back to ERA-Interim reanalyses and the simulations are run for around 20 years, from 1/9/1988 to 1/1/2009.

Figure 14(a,b) shows the climatological differences in 850 hPa winds and precipitation between the Control and Nudged experiments during JJA, for the "Philippines" and "Indonesia" regions. These results suggest that the "Indonesia" region promotes westerly anomalies extending from the South Asian monsoon westerly jet across the Philippines into the western Pacific, while the "Philippines" region promotes additional acceleration of these westerly winds as part of an anomalous cyclonic circulation that includes north-easterly anomalies over southern China. Both regions promote excess rainfall over the eastern SCS and the western Pacific. Figure 14(c,d) shows the results for the SASM and MC regions. These suggest that errors arising locally over the SASM region are directly responsible for the anticyclonic anomaly and deficit in rainfall over India and for much of the error pattern in rainfall over the equatorial Indian Ocean. The SASM region also promotes the acceleration of the westerly winds across the SCS into the western Pacific and the positive error in rainfall in those regions. The Maritime Continent region as a whole promotes acceleration of the westerly winds and increased rainfall across the SCS and the western Pacific, and an anticyclonic anomaly that represents weakening and eastward displacement of the WNPSH region. The influence of the Maritime Continent region, and particularly the Indonesian islands, in promoting the

southeasterly(easterly) wind anomalies in the eastern(central) Indian Ocean, as suggested in Sect. 3.2 and 3.3, is also confirmed by these results.

This analysis suggests that there are both local and remote contributors to the ASM errors seen in the MetUM model simulations. The experiments indicate that Indonesia and the oceans around the Philippines play a separate, but interacting, role in the development of these errors during the seasonal transition towards the Asian summer monsoon. The SASM region helps to reinforce those errors while also developing the majority of its circulation and rainfall errors locally.

### 3.5.2 Initialised simulations

The "nudging" methodology can also be applied in initialised simulations and used to track the influence of a particular region on the development of errors elsewhere. We show here, as an example of this methodology, the influence of the "Philippines" (PHL) region (used in Sect. 3.5.1) on the growth of remotely forced model systematic errors over China, the western Pacific and the Maritime Continent, over the first 15-days of NWP-2016 atmosphere-only simulations (see Table 1) conducted during June–August 2016 (Fig. 15). Consistent with the analysis of GloSea5 coupled model hindcasts in Fig. 6, the total mean error (forecast minus analysis) in the surface wind for forecast days 1, 5, and 15 (see Fig. 15(a–c)) shows the gradual emergence of the systematic errors. This includes erroneous equatorial easterlies west of Sumatra, extending to 80°E, and a large error in the western Pacific, east of the Philippines that extends north to the sub-tropics in an erroneous cyclonic pattern that reflects the weakening of the WNPSH. Other surface-wind errors are shown in the Maritime Continent, the Bay of Bengal and the western equatorial Indian Ocean off the African coast.

On day 1, the contribution of the PHL to the total error is very small, mostly confined to the PHL region as expected, but by day 5 of the NWP forecasts the PHL errors are responsible for forcing mean errors beyond the nudged region, such as the erroneous cyclonic wind in the Western Pacific subtropics, as well as errors in the Maritime Continent. These errors are consolidated by day 15 of the forecast (Fig. 15 (d–f)). For completeness, we also show the contribution to the total error from the areas outside the PHL nudging domain (Fig. 15(g–i)). A smaller area of erroneous cyclonic circulation in the Pacific occurs just south of Japan by day 5, that indicates that the systematic error in the WNPSH also has extra-tropical origins. Other wind errors not forced by the PHL region include the easterlies west of Sumatra and errors in the Bay of Bengal and the Western equatorial Indian Ocean off the African coast.

Similar experiments have been carried out with the other regions identified in Sect. 3.5.1. These also confirm that the local and remote contributions from those regions to the circulation errors in the ASM emerge in the first 5-15 days of the forecasts. These experiments illustrate the important role played by certain regions in the development, from an early stage, of systematic errors in the ASM. Future work will include applying the nudging technique to GloSea5 hindcasts in order to trace the influence of specific regions on the development of errors on seasonal timescales. Identifying such key regions provides a focus for future process analysis, model development and evaluation which may ultimately improve the model forecasts for the ASM as a whole.

## 4 Summary

We have demonstrated the application of a hierarchical approach to investigating the sources of systematic errors in the Asian summer monsoon. A range of configurations within the Met Office's seamless modelling system has been used to study the evolution of errors, separate their local and remote contributions, analyse the role of model resolution and atmosphere-ocean coupling and to start to identify processes requiring attention. While not exhaustive, this work paves the way for further, targeted, process analysis and sensitivity tests as part of future model development. A flow diagram summarising this approach is shown in Figure 16.

Our analysis suggests that there are both local and remote contributors to the ASM errors seen in the MetUM model simulations. The experiments indicate that Indonesia and the oceans around the Philippines play a separate, but interacting, role in the development of errors in the EASM, while in the SASM region the errors appear to be mainly driven locally. Furthermore, the errors in the SASM region help to reinforce the errors in the EASM. Although many of the same systematic error patterns have been found in atmosphere-only simulations (e.g. Rodriguez and Milton, 2019), SST errors also contribute, both at initialisation and through their development in a coupled response to the circulation and rainfall errors.

The Equatorial Indian Ocean develops a southeast (dry) – northwest (wet) rainfall error pattern and an east (cold) – west (warm) SST error pattern. These originate from a negative rainfall error and divergent anomalies over the Maritime Continent and a positive rainfall error and convergent anomalies over the western/central equatorial Indian Ocean, the latter being accompanied by an anticyclonic error and deficient rainfall over the Indian region. The anticyclonic error over India (which develops rapidly after initialisation) is associated with a weakening of the monsoon trough and a reduction in the number, and rainfall contribution from, monsoon lows and depressions, which also are unable to progress across northern India. This, combined with excessive rainfall over the steep orography of the eastern Himalaya that promotes convergence from the south and drying over the head of the Bay of Bengal, results in reduced rainfall over the Bay while that over the Myanmar orography is increased, with an associated acceleration of the westerly flow across the Bay of Bengal and southeast Asia into the South China Sea. This converges with the southerly anomalies from the Maritime Continent region, promoting further rainfall and creating a positive feedback that develops a westerly wind error (extension of the westerly jet) across the SCS and the Philippines into the western Pacific. The SSTs in the EEIO and in the SCS respond to these changes by (ultimately) cooling. In the EEIO, this is exacerbated by an ocean mixed layer that is too shallow.

While further analysis is needed to investigate the processes involved and how they are mis-represented in the models, we have narrowed down some of the regions responsible which will allow us to target future detailed investigations. We have identified particular model errors whose origins lie clearly in the atmospheric component, while other errors appear to have an origin in the ocean. Coupled feedbacks exacerbate such errors and also make it difficult to unambiguously identify misrepresentation of either atmosphere or ocean processes. In addition, biases over land and ocean can evolve differently, and this will modify the land-sea temperature contrast with a possible impact on the ASM (e.g. Chen and Bordoni, 2016; Lutsko et al., 2019). The nudging technique, applied separately over land and sea points, could shed further light on the role

of errors in land-sea temperature contrast. This will be explored in future work, as well as applying nudging of the ocean

model separately from, and together with, the atmosphere.

We have also shown that the development of the errors in the first few weeks depends on when the hindcasts are initialised in relation to the broadscale monsoon transition that typically occurs in mid-May. This is evident in the EEIO and the SCS, and also in the Arabian Sea and northern Bay of Bengal. This may have implications for monsoon forecasting on short and medium-range timescales, particularly when coupled NWP models are used. In future work we will use sensitivity

experiments to explore the separate and interacting role of atmosphere and ocean in the development of errors in each of these regions. Finally, consistent with previous studies using this model (e.g. Johnson et al, 2016), we find that these systematic errors and their development are largely insensitive to changes in horizontal resolution, despite the improved representation of orography and coastlines in the higher resolution models.

## 5 Conclusions

In this study we have demonstrated the use of a range of modelling tools and techniques aimed at understanding the sources of error in monsoon regions, using the specific example of the ASM errors in the MetUM model. The tools and techniques allow close examination of the error development after initialization, the separation of the roles of local processes and remote teleconnections, the identification of the contribution from errors developing in particular regions to the ASM error as a whole, and understanding of the role of atmosphere-ocean coupling. While there have been several works that use

initialized modelling frameworks to diagnose the origins of systematic errors in the Asian summer monsoon (such as those referenced in the Introduction), the use of a variety of techniques such as those described here that includes both coupled and atmosphere-only configurations and regional modelling to analyse the development and sources of particular errors on a range of timescales has not, to our knowledge, been demonstrated.

This analysis methodology benefits from the use of a seamless modelling system, where different configurations of a

570 modelling system that are used for forecasting on different timescales share very similar physical and dynamical formulations. This allows the development of systematic errors to be studied on a range of timescales, and the roles of resolution and ocean-atmosphere coupling to be studied, without the complication of different physical parameterizations or dynamical cores that other multi-model studies might include. This approach also allows the whole suite of models to benefit from improvements that ultimately result from better understanding of the errors and informed, targeted, model

development.

Our study highlights a number of different techniques that can be employed to investigate the sources of model error in a particular region. Once these are known, further work can be done to explore the local processes contributing to this behaviour and their sensitivity to changes in physical parameterizations in the model. While further work is clearly necessary, we hope that this work inspires other modelling groups to carry out similar analysis with their own models in

order that some of the major, long-lasting, systematic errors in GCMs can ultimately be reduced.

## Code and data availability

Due to intellectual property right restrictions, we cannot provide either the source code or documentation papers for the Met Office Unified Model (MetUM). The MetUM is available for use under licence. For further information on how to apply for a licence see https://www.metoffice.gov.uk/research/approach/collaboration/unified-model/partnership (last access: 585 16 July 2020). JULES is available under licence free of charge. For further information on how to gain permission to use JULES for research purposes, see https://jules.jchmr.org/software-and-documentation. The model code for NEMO v3.4 is available from the NEMO website (www.nemo-ocean.eu). The model code for CICE is freely available from the United States Los Alamos National Laboratory (http://oceans11.lanl.gov/trac/CICE/wiki/SourceCode). Model data used in this study are archived at the Met Office, and 590 are available to research collaborators upon request.

## Author contribution

Gill Martin initiated the study and carried out the analysis of the seasonal hindcasts and free-running climate simulations. Richard Levine designed, ran and analysed the RCM experiments. José Rodriguez and Michael Vellinga analysed the operational coupled and uncoupled NWP simulations. José Rodriguez designed, ran and analysed the nudged simulations. 595 All authors contributed to the writing of the manuscript.

## Competing interests

The authors declare that they have no conflict of interest.

## Acknowledgements

This work and its contributors were supported by the UK-China Research & Innovation Partnership Fund through the Met 600 Office Climate Science for Service Partnership (CSSP) China as part of the Newton Fund, and by the Weather and Climate Science for Service Partnership (WCSSP) India, a collaborative initiative between the Met Office, supported by the UK Government's Newton Fund, and the Indian Ministry of Earth Sciences (MoES). NOAA High Resolution SST data were provided by the NOAA/OAR/ESRL PSD, Boulder, Colorado, USA, from their Web site at https://www.esrl.noaa.gov/psd/. ERA5 data were obtained from the Copernicus Climate Change Service Climate Data Store at 605 https://cds.climate.copernicus.eu/cdsapp#!/home.

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

**Table 1: Model configurations used in this study.**

| Configuration | Atmosphere Resolution (longitude x latitude) | Ocean resolution (longitude x latitude) | Notes |
|---|---|---|---|
| GC2.0 | 0.833° x 0.556° | ORCA025 tripolar grid: 0.25° x 0.25° between 80S-80N | 30-year coupled climate simulation forced by perpetual present-day greenhouse gases and aerosols (details in Williams et al., 2015). GC2.0 is comprised of GA6.0 & GL6.0 (Walters et al., 2017), GO5.0 (Megann et al., 2014) and GSI6.0 (Rae et al., 2015). |
| GloSea5-GC2 | 0.833° x 0.556° | ORCA025 tripolar grid: 0.25° x 0.25° between 80S-80N | 23-year hindcast ensemble (1993-2016) from operational long-range forecast system (MacLachlan et al, 2015). |
| RCM GA7.0 | 0.44° × 0.44° | N/A | Regional climate model forced at lateral boundaries by 6-hourly ERA-I reanalyses from 1989-2008. Domains shown in Fig. 3, each uses rotated north pole at 61°N, 296.3°E. |
| NWP GA6.1 hindcasts (denoted UNCPLDNWP) | 0.234° x 0.156° | N/A | 7-day hindcasts initialised once per day through JJA of 2016. GA6.1 includes a small number of scientific differences from GA6.0 (see Walters et al., 2017). |
| NWP GC2.0 hindcasts (denoted CPLDNWP) | 0.234° x 0.156° | ORCA025 tripolar grid: 0.25° x 0.25° between 80S-80N | 15-day hindcasts initialised once per day through JJA of 2016 (Vellinga et al., 2020) |
| AGCM GA7.0 (denoted AGCM-N216) | 0.833° x 0.556° | N/A | 20-year atmosphere-only climate run forced by observed SSTs, 1989-2008. GA7.0 is described by Walters et al. (2019). |
| GA7.0 relaxation experiments (denoted AGCM-N96) | 1.88° x 1.25° | N/A | 20-year atmosphere-only climate runs forced by observed SSTs, 1989-2008, relaxed to ERA-I with a 6-hourly relaxation time scale within specific regions (shown in Fig. 14), denoted "Nudged", compared with Control at same resolution. |
| NWP GA6.1 hindcast relaxation experiments (denoted NWP-2016) | 0.833° x 0.556° | N/A | 15-day hindcasts initialised once per day through JJA of 2016, relaxed to ERA-I with a 6-hourly relaxation time scale within specific regions, compared with Control at same resolution. |

**Table 2: Summary of the errors developing within the China1 domain of the RCM, and the effects of extending the domain boundaries on either side of the China1 domain compared with the effects of China1 itself, as shown in Figure 5.**

| | **China1N** (includes China1):<br>Some additional southwesterly anomalies and drying over southern China. | |
|---|---|---|
| **China1W** (includes China1):<br>Some additional dry rainfall anomalies over southern China. Increase in India dry rainfall anomalies. | **China1:**<br>Southerly and south-westerly anomalies over southeast China, the South China Sea and the Philippine Sea.<br>More widespread wet error than the AGCM over southern steep orographic edges of Tibetan Plateau, and most of southern China.<br>The circulation anomalies over southeast China and the surrounding seas in China1 are opposite to those in the AGCM. | **China1E** (includes China1):<br>Anomalous low-level easterly wind component towards eastern China as part of weakening and northward displacement of the WNPSH.<br>Associated increase in rainfall over land and decreased rainfall offshore. |
| **China1SW** (incl. China1, China1S, China1W):<br>On top of China1S and China1W impacts, further increase in India dry rainfall anomalies due to inclusion of more areas that contribute to AGCM dry India bias, and additional drying over southern and eastern China. | **China1S** (includes China1):<br>Weak contributions to westerly anomalies across the South China Sea and to the easterly anomalies and drying over the Yangtze River Basin. | **China1SE** (includes China1, China1S, China1E):<br>Contributes north-easterly anomalies (not present in China1E and China1S) as WNPSH is further weakened and displaced.<br>Additional drying over the middle/lower Yangtze River Basin. |

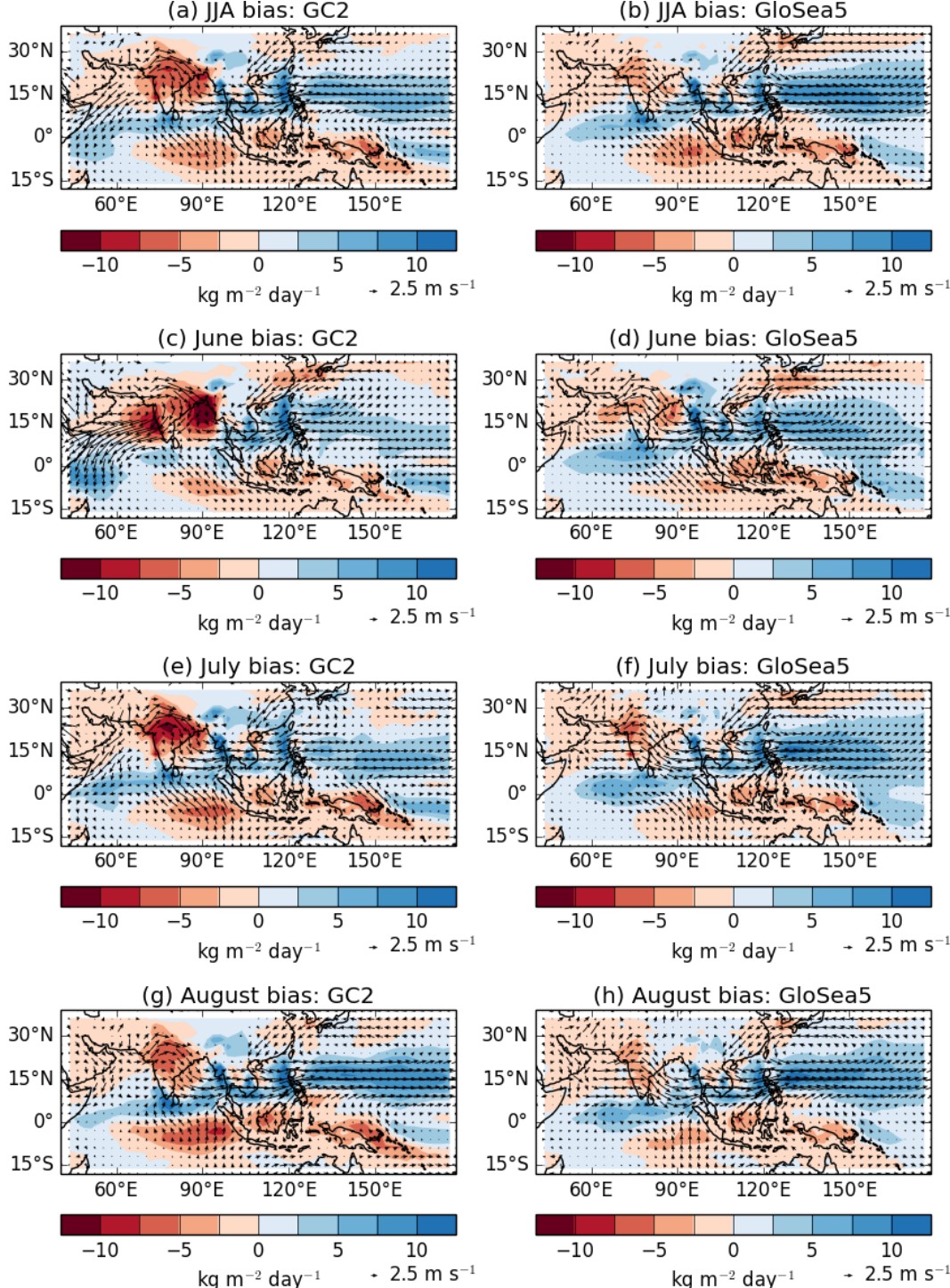

**Figure 1:** (a,b) JJA, (c,d) June, (e,f) July and (g,h) August climatological errors in precipitation (against GPCP observations) and 850 hPa winds (against ERA-Interim reanalyses) from the current GloSea5 23-year operational hindcast ensemble initialised each year on four start dates (1, 9, 17, 25) in April, May, June and July respectively, with 7 members per start date.

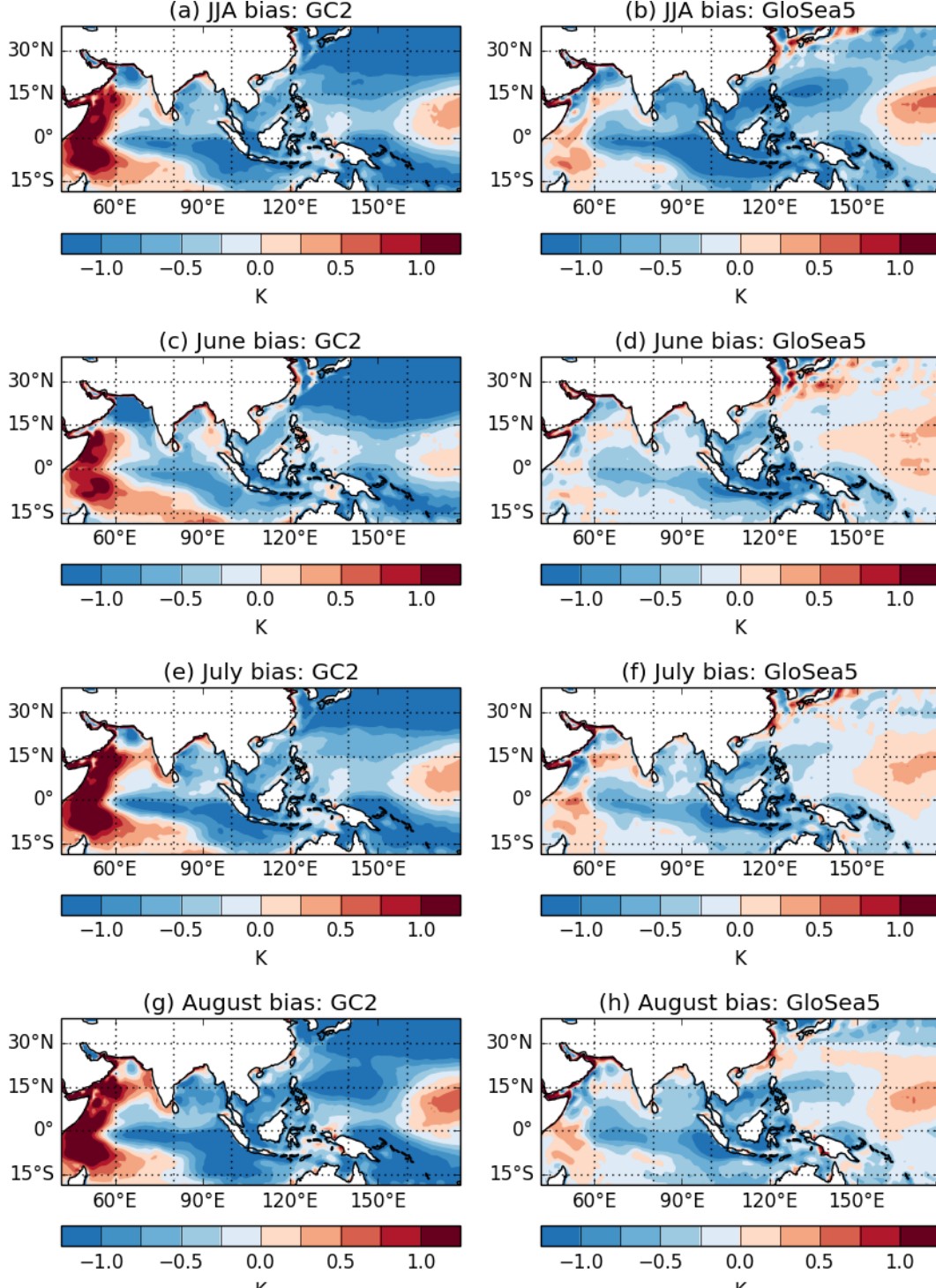

Figure 2: As Fig 1 but for sea surface temperature, compared against OISSTv2 observations.

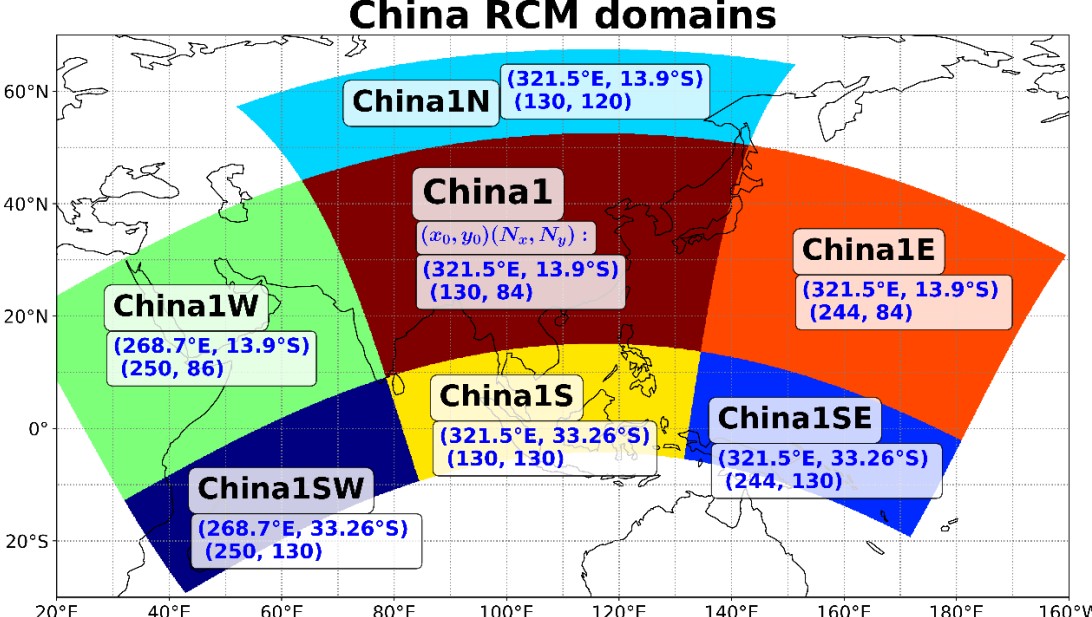

**China RCM domains**

China1N — (321.5°E, 13.9°S) (130, 120)

China1 — $(x_0, y_0)(N_x, N_y)$ : (321.5°E, 13.9°S) (130, 84)

China1E — (321.5°E, 13.9°S) (244, 84)

China1W — (268.7°E, 13.9°S) (250, 86)

China1S — (321.5°E, 33.26°S) (130, 130)

China1SE — (321.5°E, 33.26°S) (244, 130)

China1SW — (268.7°E, 33.26°S) (250, 130)

**Figure 3: Domains used in Regional Climate Model experiments. China1 is the smallest, central domain, with the other domains obtained by extending this to the north, south, east and west. All RCM simulations use a 0.44° × 0.44° resolution grid and a rotated pole at 61°N, 296.3°E. Coordinates for each region are shown in the form $(x_0, y_0)(N_x, N_y)$ where $(x_0, y_0)$ is the position of the lower left hand corner of the region (in rotated pole coordinates) and $(N_x, N_y)$ is the number of grid points in the $x$ and $y$ direction. China1 is the central (and smallest) domain, and is included in all other domains. China1W, China1E, China1S, China1N are extensions of China1 to the west, east, south and north respectively. China1SW overlaps with the China1S and China1W (and China1) regions, extending into both south and west directions. China1SE overlaps with the China1S and China1E (and China1) regions, extending into both south and east directions.**

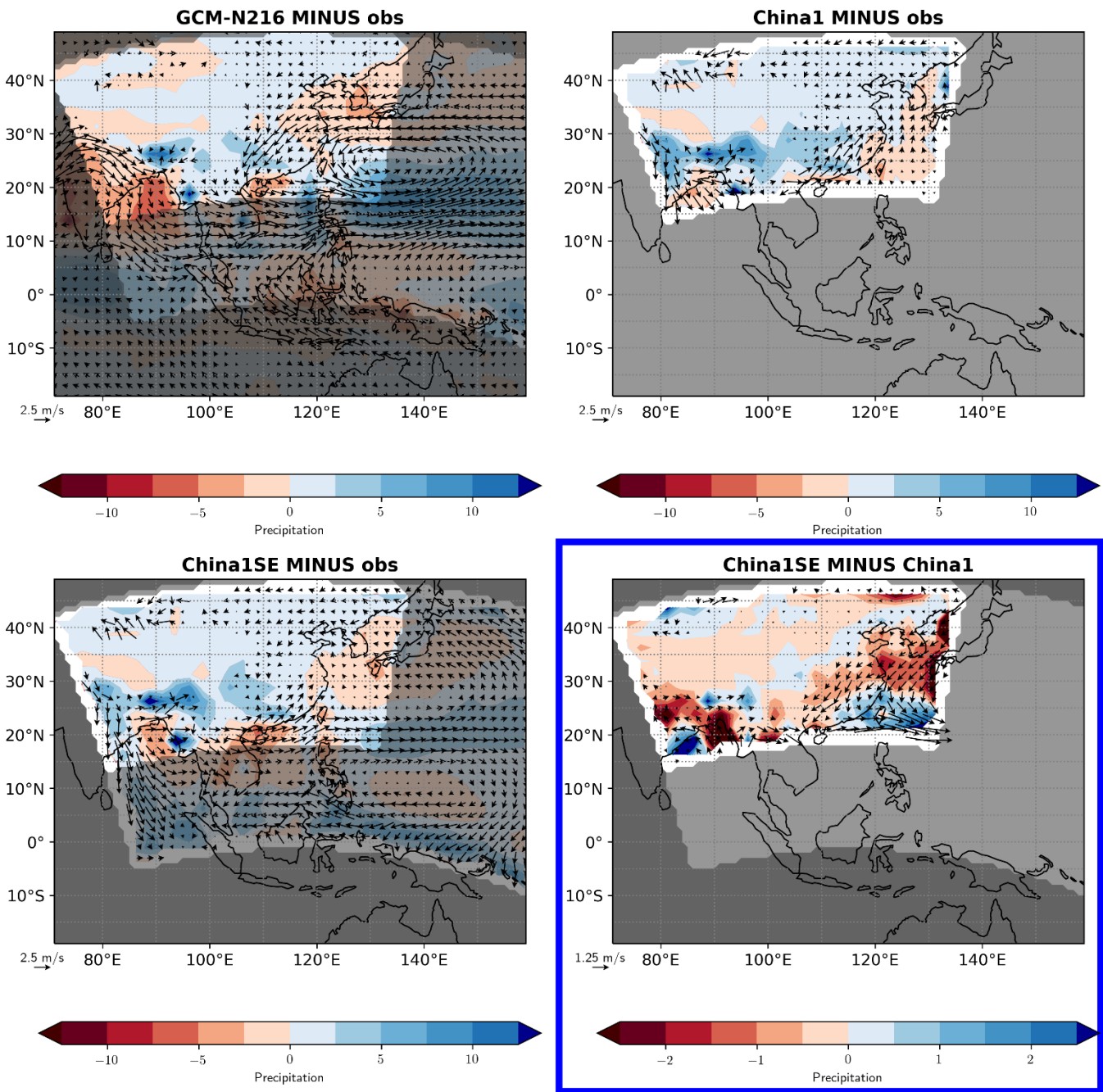

**Figure 4: (top)** JJA climatological (1989-2008) precipitation and 850hPa wind errors in the AGCM-N216 simulation and the China1 RCM simulation vs TRMM and ERA-Interim (obs). **(bottom)** Effects of extending the domain towards the south and east: **(left)** China1SE minus obs; **(right)** China1SE minus China1. Colour bar indicates precipitation differences (mm/day) and vectors indicate differences in 850 hPa winds. NOTE difference in contour intervals and vector scale for lower right panel. Non-shaded area highlights China1 region, lightly shaded area highlights China1SE region, darker shaded area highlights areas not covered by China1 and China1SE domains.

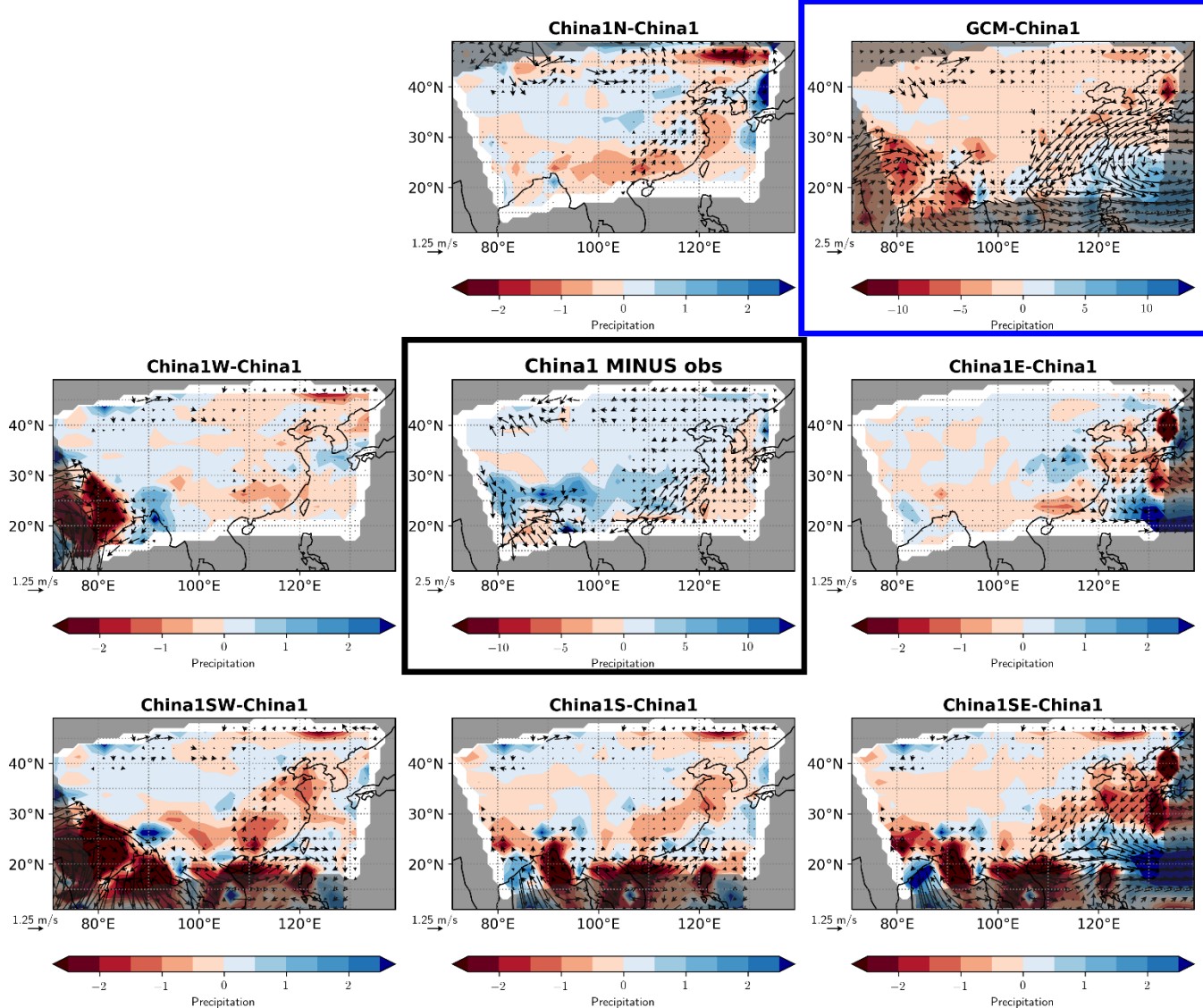

**Figure 5: Effects of extending the China1 domain in different directions, as indicated in Fig. 3. Differences are against China1 in each case (except outside the China1 region, where they are differences from observations), while the centre panel shows China1 minus observations. NOTE the differences in contour interval and vector scale between the centre and top right panels and the other panels. Plots are for JJA 1989-2008. Colour bar indicates precipitation differences (mm/day) and vectors indicate differences in 850 hPa winds.**

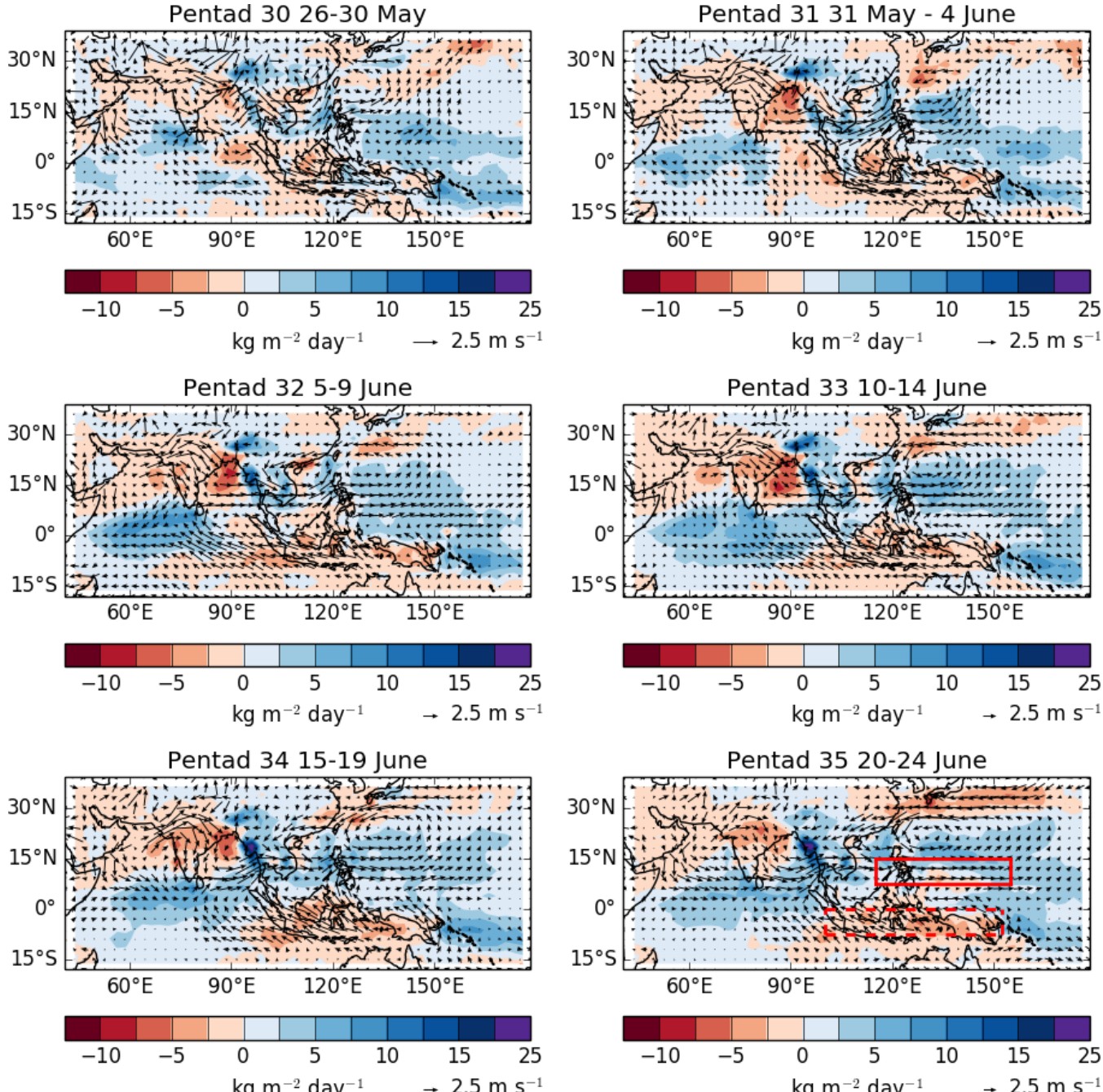

**Figure 6: Development of climatological errors in 850 hPa winds and precipitation (against ERA-I and GPCPv2), pentad by pentad after initialisation on 25th May. The solid red box indicates the "Philippines" region used Fig. 8(c) and in the nudging experiments. The red dashed box shows the "Indonesia" nudging region.**

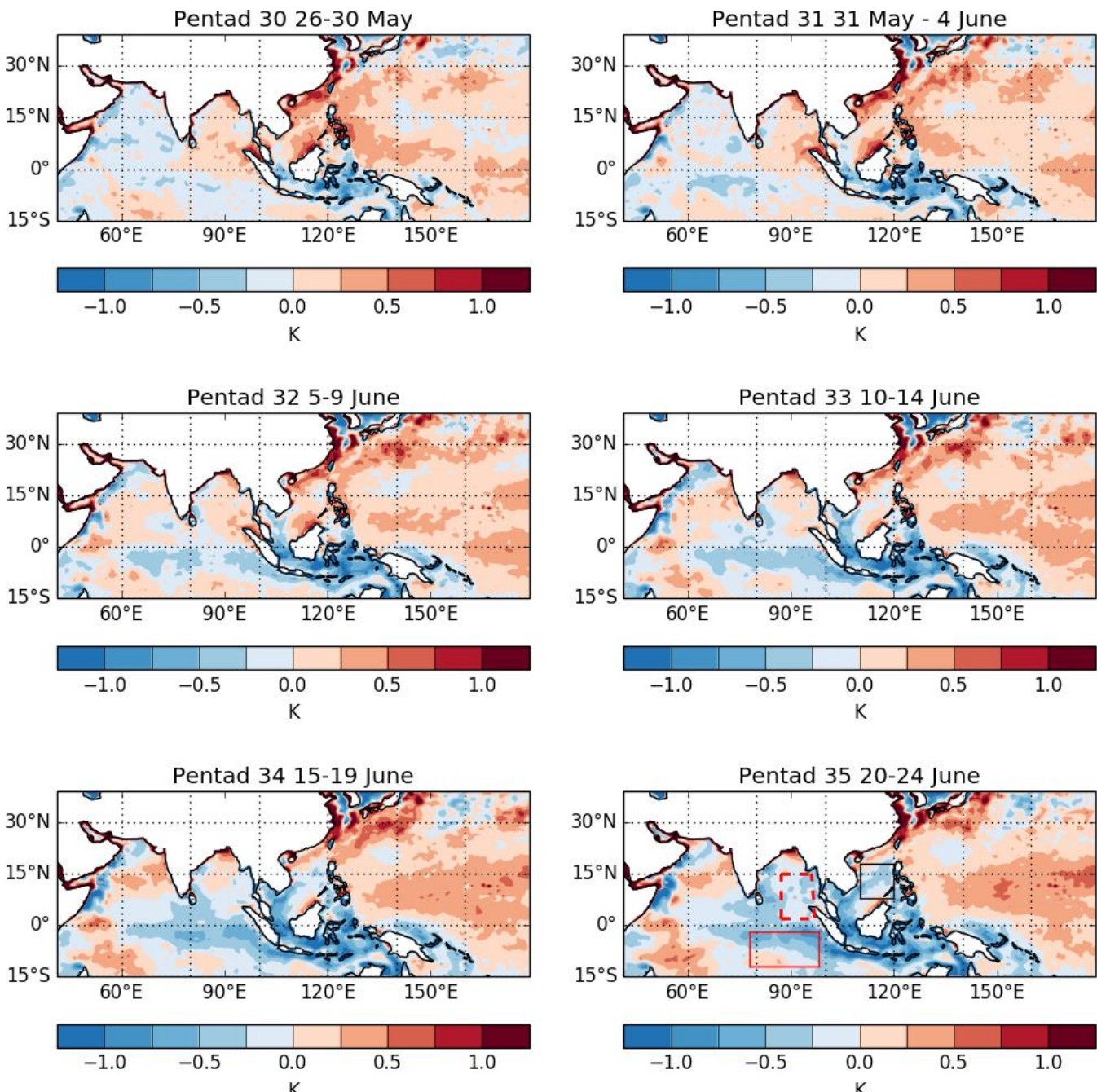

**Figure 7: Development of climatological errors in SST (OISSTv2), pentad by pentad after initialisation on 25th May. The solid and dashed red and black boxes indicate the southern and northern EEIO and SCS boxes used in Fig. 8, respectively.**

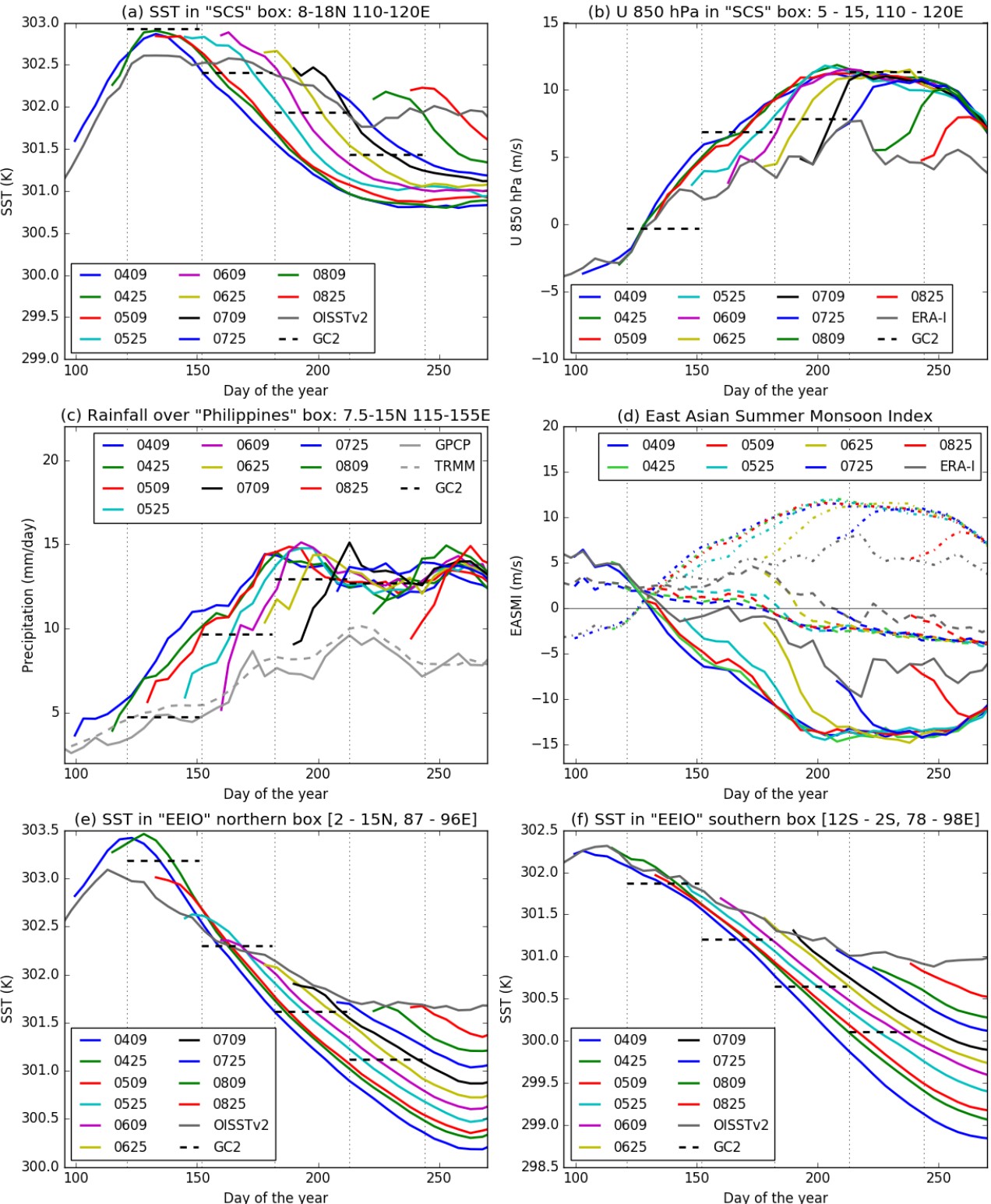

**Figure 8: Climatological spin-up of various quantities over key regions in GloSea5 hindcast ensembles initialised on 9, 25 April (0409, 0425), 9, 25 May (0509, 0525), 9, 25 June (0609, 0625), 9, 25 July (0709, 0725), 9, 25 August (0809, 0825), 7 members for each, averaged over 1993-2015: (a) SST in South China Sea (8°-18°N, 110°-120°E); (b) 850 hPa zonal wind over SCS box used by Wang et al (2004) (5°-15°N, 110°-120°E); (c) Rainfall over the Philippines box (7.5°-15°N, 115°-155°E); (d) EASMI: 850 hPa zonal wind difference (22.5°-32.5°N, 110°-140°E) – (5°-15°N, 90°-130°E); (e) SST in EEIO northern box (2°-15°N, 87°-96°E); (f) SST in EEIO southern box (12°S-2°S, 78°-98°E). Pentad SST is shown from OISSTv2, pentad rainfall from GPCP and TRMM observations, and EASMI from ERA-I. For EASMI, the solid lines indicate the index, while the dashed line indicates the northern box, dot-dash indicates southern box. Black dashed horizontal lines indicate monthly averages from free-running GC2 coupled model simulations. Day of the year is calculated using the Gregorian calendar (without leap years); day 121 corresponds to 1 May. Vertical dotted lines indicate the start of each month.**

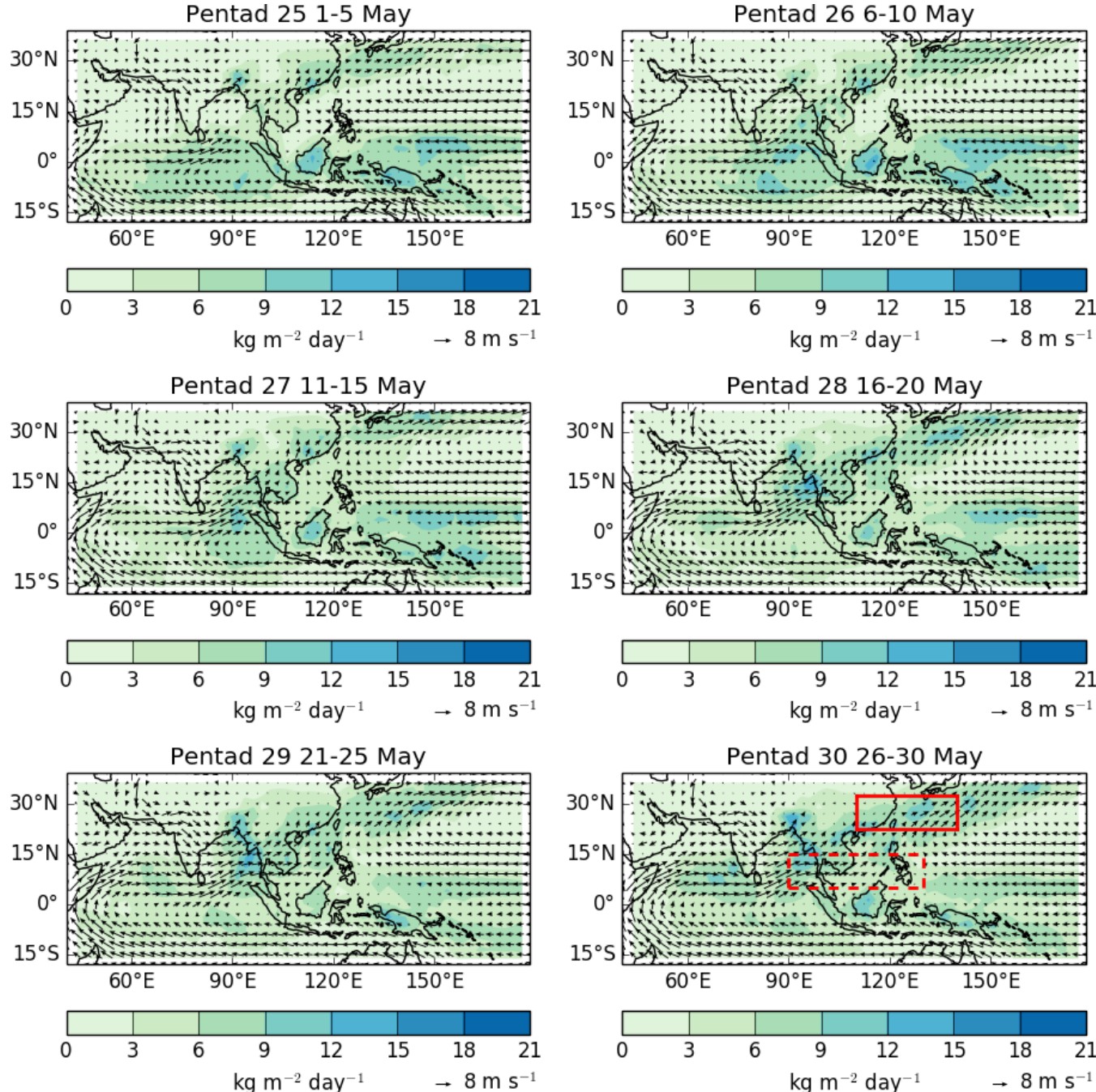

**Figure 9: Pentad evolution of 850 hPa winds (ERA-Interim) and rainfall (GPCPv2) through the broadscale seasonal transition towards the Asian summer monsoon season, heralded by the reversal of wind direction over the South China Sea which typically occurs around pentad 28 (Wang et al., 2004). The red boxes on the final panel indicate the northern (solid) and southern (dashed) regions that are used in the calculation of the East Asian Summer Monsoon Index (see Wang et al. (2008) and Fig. 8d).**

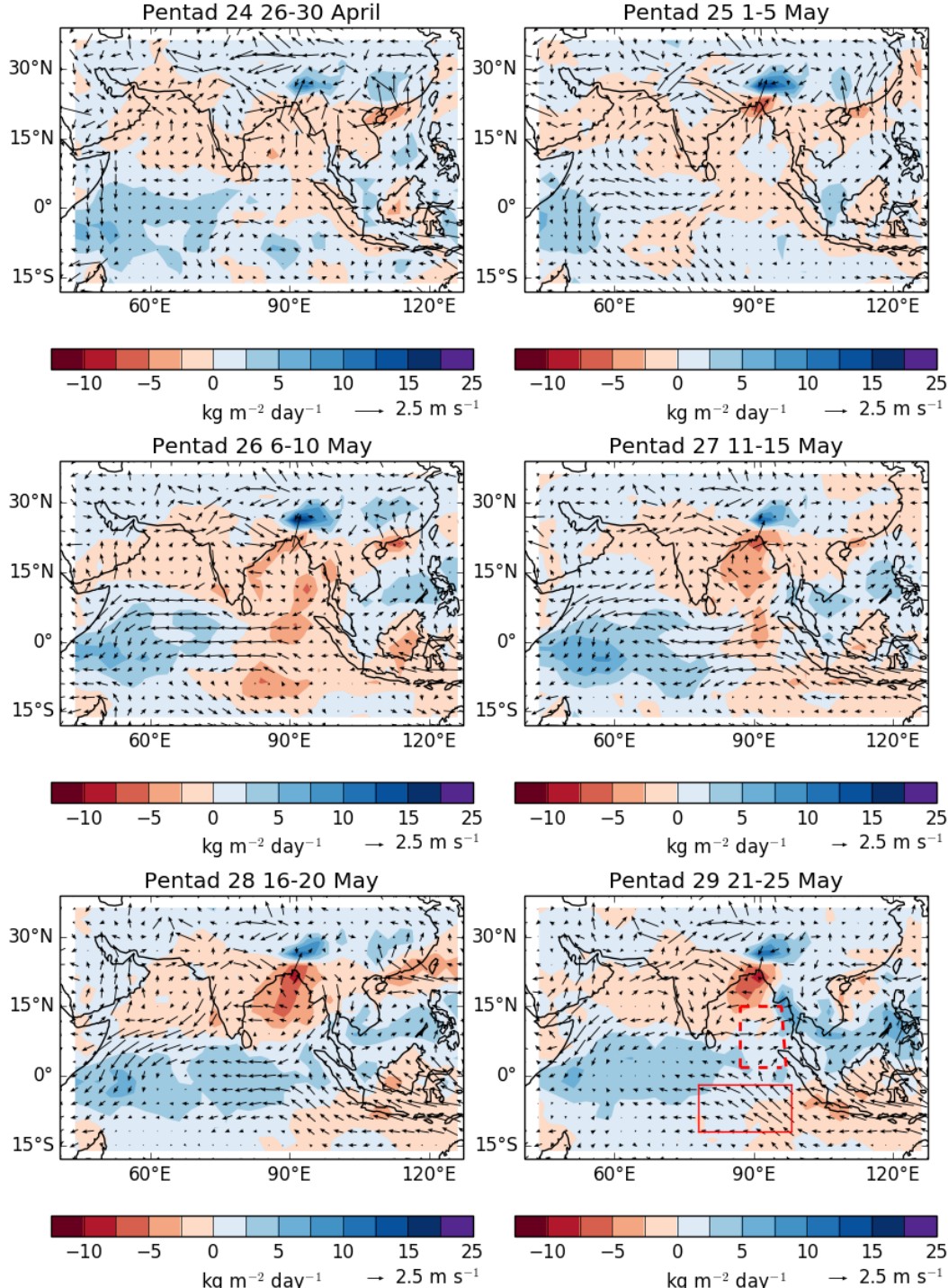

**Figure 10: Development of climatological errors in 850 hPa winds and precipitation (against ERA-I and GPCPv2), pentad by pentad after initialisation on 25th April. The red boxes indicate the northern (dashed) and southern (solid) EEIO boxes used in Fig. 8(e, f).**

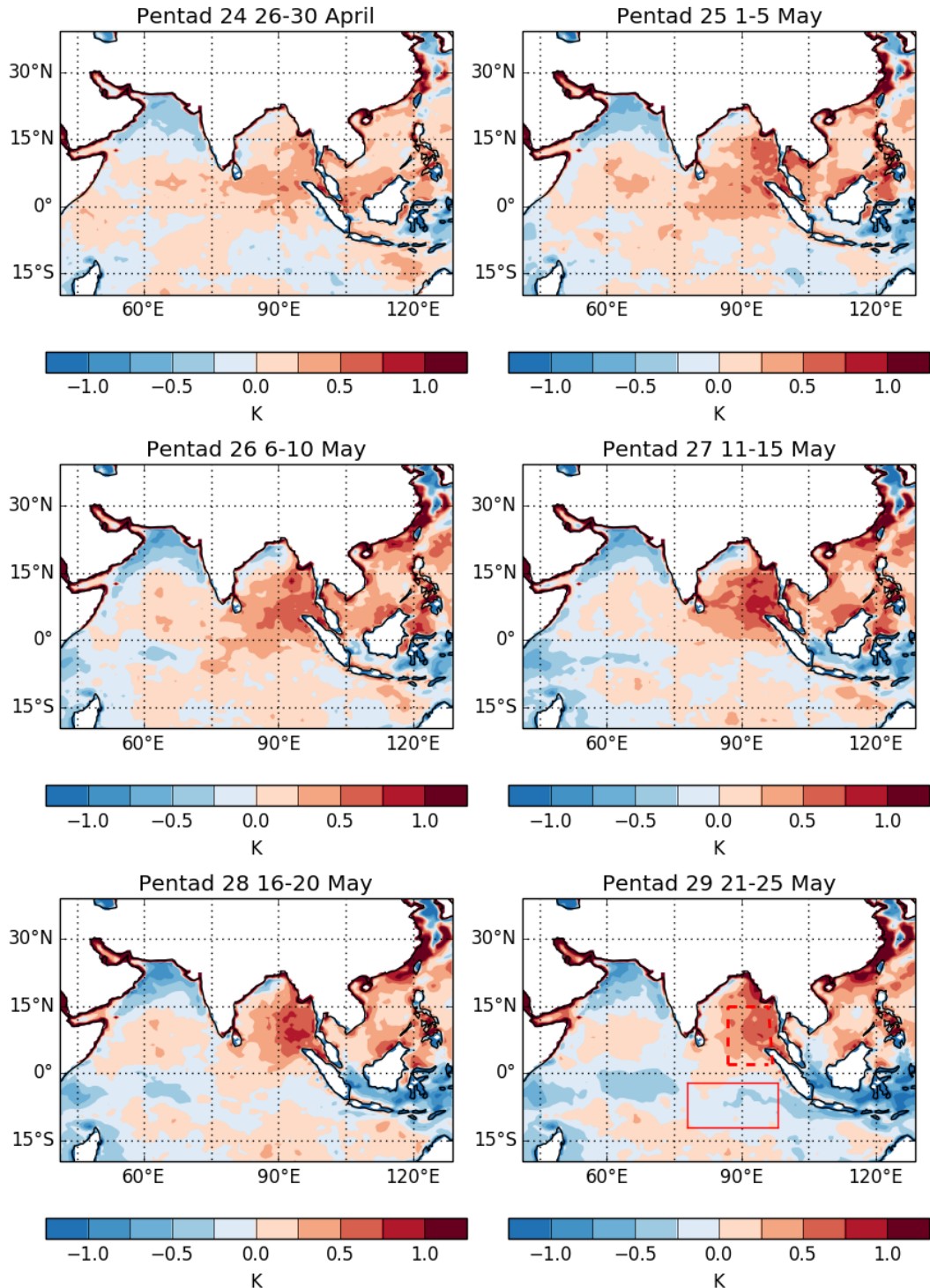

**Figure 11: Development of climatological errors in SST (OISSTv2), pentad by pentad after initialisation on 25[th] April. The red boxes indicate the northern (dashed) and southern (solid) EEIO boxes used in Fig. 8(e, f).**

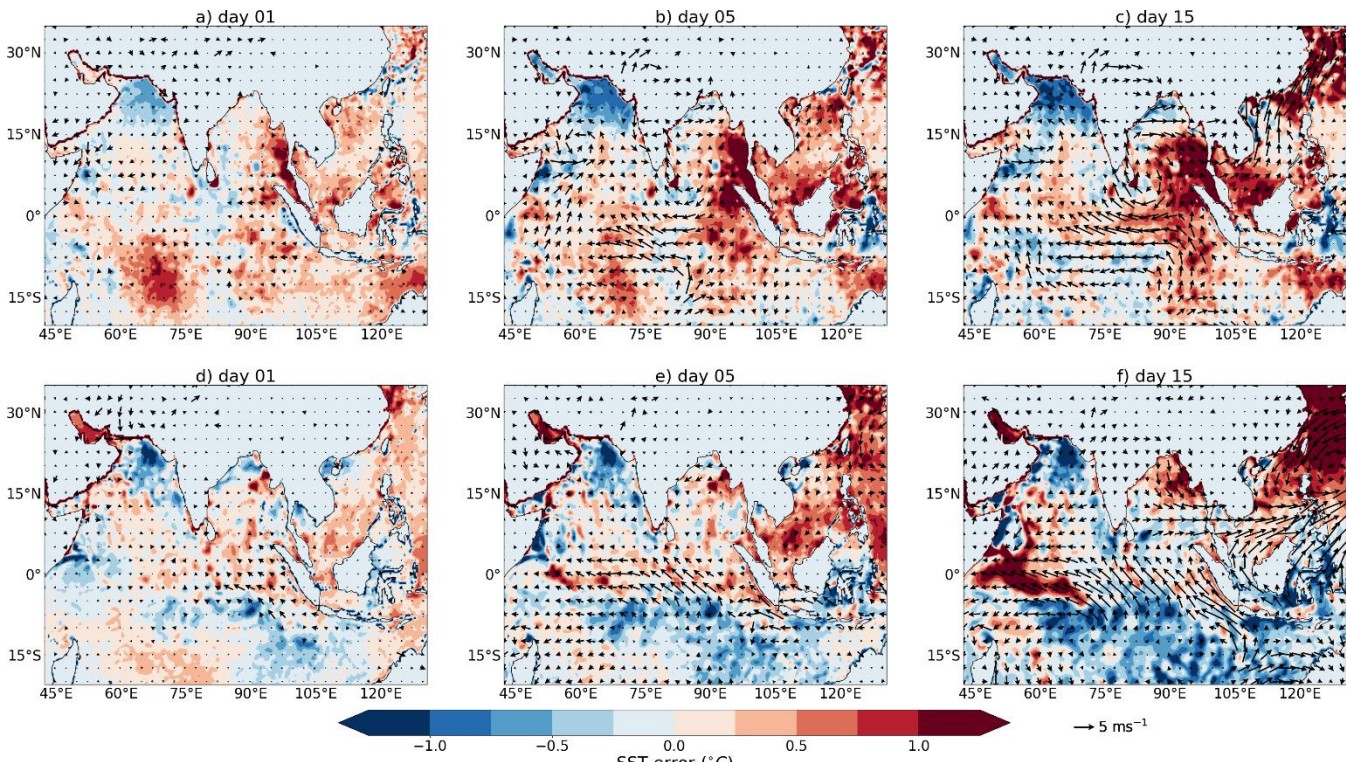

**Figure 12:** SST error (colours, °C) and 10-m wind errors (arrows) from CPLDNWP simulations with respect to OISSTv2 in composites before (10-19 May; top row) and after (10 – 23 July; bottom row) the broadscale seasonal transition in 2016, for forecast lead times of 1, 5 and 15 days.

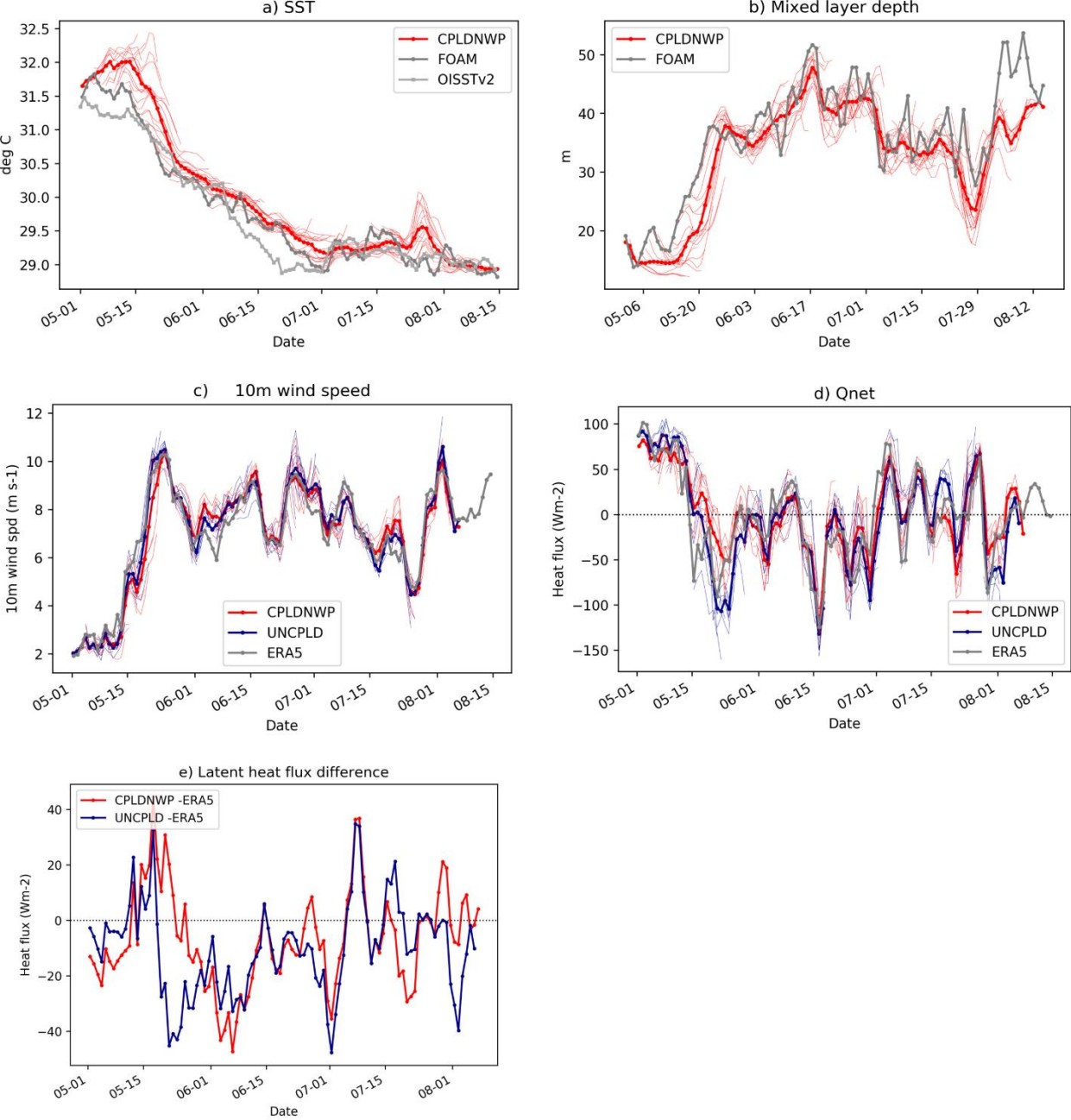

**Figure 13: Time series of daily forecasts initialised between 1 May–31 July 2016. Individual forecasts (thin lines) are averaged at each validity time ('Date') into a mean forecast (coloured heavy line). Grey lines are from analyses. Lines show area-averages over EEIO northern box (2°-15°N, 87°-96°E; see Fig. 8e) (a) SST from CPLDNWP and FOAM analyses, out to day 15; (b) mixed layer depth calculated using daily mean temperature and salinity from CPLDNWP and FOAM analyses, out to day 15; (c) 10m wind**
**speed from CPLDNWP (red) and UNCPLDNWP (blue) out to day 7; (d) net surface heat flux from CPLDNWP (red) and UNCPLDNWP (blue) out to day 7. Positive (negative) value denotes net flux into (out of) the ocean; (e) Mean forecast error of latent heat flux in CPLDNWP and UNCPLDNWP forecasts, relative to ERA5.**

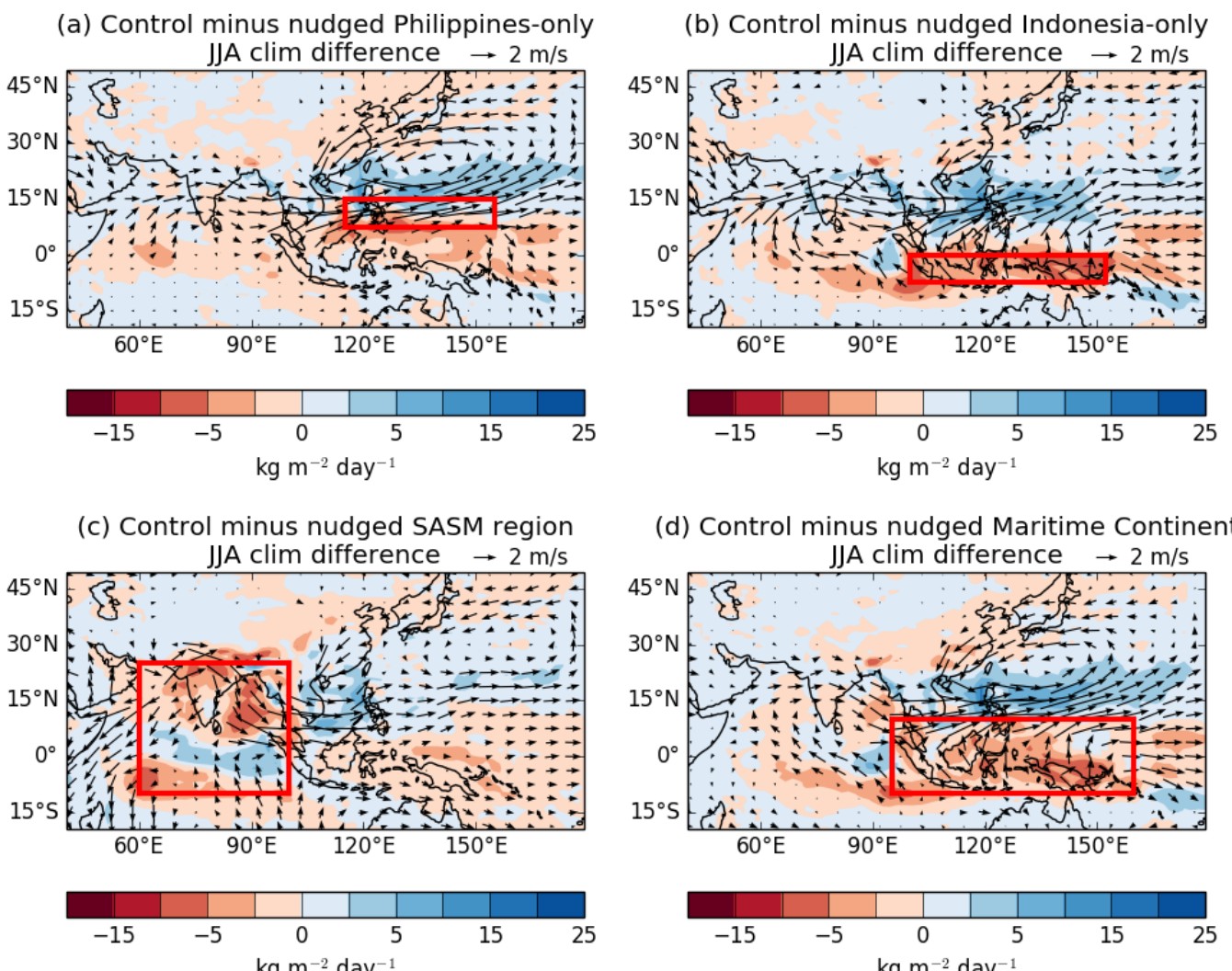

**Figure 14: Climatological differences in 850 hPa winds and rainfall in JJA between "Nudged" experiments and their Control,**
**indicating the influence on the Control of errors developing in four different regions used for the Nudged experiments: (a)**
**Philippines; (b) Indonesia; (c) South Asian summer monsoon (SASM); (d) Maritime Continent. The nudged regions are shown in**
**red: "Philippines" [115°-155°E, 7.5°-15°N], "Indonesia" [100°-152.5°E, 7.5°S-0°N], "SASM" [60° - 100°E, 10°S - 25°N], and**
**"Maritime Continent" [95° - 160°E, 10°S - 10°N].**

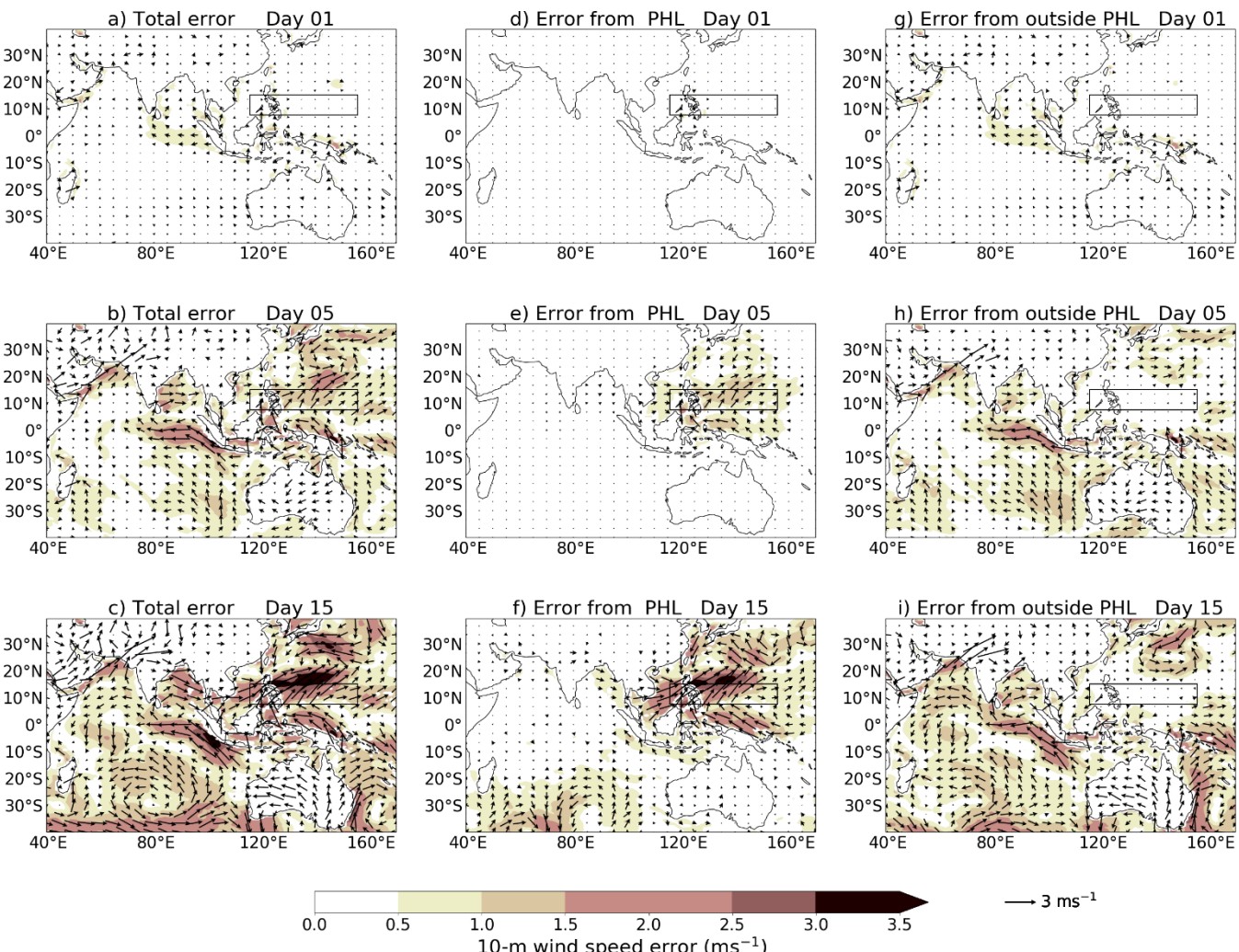

**Figure 15: June–August 2016 mean 10m-wind errors (arrows) and their magnitudes (colours, m s⁻¹). (a–c) atmosphere-only NWP-2016 total errors, with respect to MetUM analysis, for forecast lead times of 1, 5 and 15 days. (d–f) NWP-2016 Control run minus Philippines nudging (nudged region shown as black box), showing errors forced from the Philippines for forecast lead times of 1, 5 and 15 days. (g–i) Philippines nudging minus global nudging, showing errors forced outside of the Philippines for forecast lead times of 1, 5 and 15 days.**

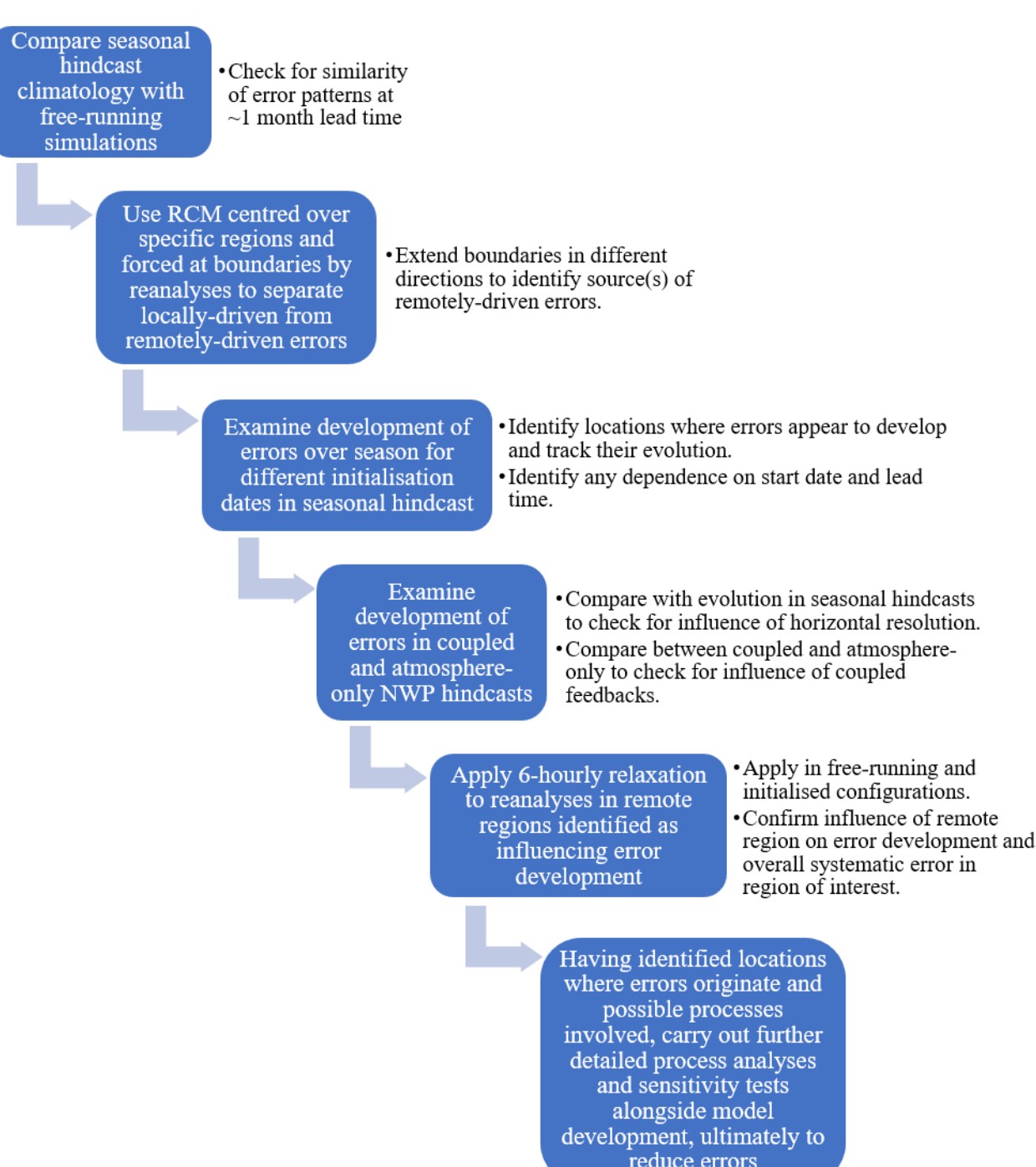

**Figure 16: Flow diagram of the hierarchical methodology applied in this study.**