# Peer review of "Understanding the development of systematic errors in the Asian Summer Monsoon"

_Geoscientific Model Development, 2020_

## Referee Comment (RC1) · Anonymous Referee #1 · 3 Nov 2020

Summary

This paper describes systematic modelling errors for the Asian Summer Monsoon on both weather and climate scales using a suite of tools including different model configurations within the MetUM, Unified modelling framework, including global, regional, nudged, and initialized prediction/hindcast techniques. It highlights the utility of the methods and tools employed, reports on various errors, and proposes sources of those errors.

Overall Comments

Using the MetUM unified modelling framework to decompose systematic modelling errors for the Asian Summer Monsoon is a wonderful example of the great utility of applying one framework to a science problem, in this case, quantifying systematic modelling errors in a monsoon system. The paper contains an enormous amount of information that will be useful to modellers to improve skill for prediction (and projections) for the EASM. Although I appreciate the challenge of presenting this work in a concise and digestible way, I feel some improvements can be made, primarily to figure organization and better descriptions in the manuscript. To help guide the reader and improve the readability, I suggest the following:

1. To help guide the reader on experimental design, I recommend use of flowcharts for modelling tools and experiment description. 2. To help summarize the regional climate modelling results in Section 3.2, consider summary table (see specific comments). 3. To help digest and follow discussion on the initialized hindcasts in Section 3.3., consider reorganizing your figure suite such that specific locations (or errors foci in the text) are highlighted (see specific comments). If this is not possible, perhaps sub-heading per error topic, and better labelling on the figures will help.

Specific Comments

Line 95: An explanation of N is needed for this grid system beyond what you have here. This will also help define the reader interpret "768" from N768 on Line 129, and N96 on Line 134.

Figure 1: What is the reference vector?

Line 115: This isn't clear to me: Did you originally run RCM with GA6.0 physics? Or is your statement on a better Indian subcontinent simulation based on the GA6 vs GA7 comparison? I am not suggesting re-running anything, just a clarification on the justification for using GA7.0 configuration rather than that what was used for the global simulations. Also, to be clear, you used GA7.0 to force RCM (and not simply using the same model configuration)?

Line 169 and Figure 2: The mean JJA cold bias of GloSea5 for parts of the Indian

Ocean, around Malaysia, and perhaps Western Pacific look larger than the individual months?

Figure 3: The caption should note where domains overlap and also describe what N1 represents.

Line 222 and Figure 4: It is hard to compare the GA7 GCM with Figure 1 top left with different color contours, scales, and vector arrows. It might be helpful to add a panel in this figure to truly compare the two. Also, it might be a good opportunity to discuss the improvements moving from GA6 to GA7 which would be interesting to readers of GMD.

Line 239: The westward extension in ChinaW seems to have a rather large impact over the Indian subcontinent. Explanation?

Section 3.2: To elucidate the local/remote implications of each domain, one suggestion would be to make a summary table, i.e. something like, one row per domain; one column for remote influence notations; one column for local influence notations.

Figure 6: What is the reference vector?

Paragraph 259 and Figure 6: Comments on the dry biases in the Bay of Bengal?

Figure 7: What are your thoughts on what is going on in the Bay of Bengal. This cannot be explained by SSTs.

Line 280: The N/S dipole seems weak.

Section 3.3: It is hard to follow specific locations for much of the discussion. I recommend picking a few key areas and designing your figures (6-9) around specific locations/error sources (i.e. one location/error per figure but include the information contained across 6 – 9 but also 10-13) This might help to clearly show the progressions and biases. For example, the South China sea area, or the Bay of Bengal, or EEIO. Full plots as shown here can be supplemental for readers interested in some-

thing the authors do not highlight, but for the discussion explicitly called out in the text, there needs to be better organization of figures.

Figures 7 and 9: Please define all components of the figures in the captions or note them in the text. I don't see an explanation of the red dashed box?

Line 306: Define SCSSM.

Figure 11d: I feel like there is much to unpack from this panel beyond the few paragraphs in the text. I see that the dashed/dashed-dotted lines are defined in the caption, but some attention to these should be paid in the text with further explanation as to interpretation.

Figure 12: Shading = color contours? What is the reference vector?

Lines 351,357,370 and Figure 13: CPLDNW, UNCPLD, and FOAM, although we can guess, should be explicitly defined.

Figure 14: Labelling and boxes should be cleaned up and consistent.

Figure 16: What is the reference vector?

---

## Referee Comment (RC2) · Anonymous Referee #2 · 5 Nov 2020

This article develops a systematic method to detect biases in model simulations to improve the representation of various features of the Asian summer monsoon system in climate models. The study used multiple configurations of the Met Office Unified Model which encompass global climate simulations (fully coupled and atmosphere only), regional climate simulations, and regional nudging simulations. The authors focused on ocean-centric regions such as the Indian Ocean, Maritime Continent, and the Philippines to demonstrate the growth of regional erroneous atmospheric-ocean circulation over time that can impact the Asian summer monsoon system.

General Comments

By providing a multilayered framework when simulating the Asian monsoon system the authors identified key ocean regions that produce systematic errors and that if corrected will improve how climate and weather models simulate the system. That alone is crucial for the field and although this framework is overall beneficial, the presentation of the text and results could be further improved.

Specific Comments

The title and abstract could use some refocusing no need to mention Asian monsoon if your main goal is only the EASM. Otherwise, the authors should add some minor additional work to fully represent the title.

The authors should restructure the manuscript into EASM, SASM, and Southeast Asia analyses.

The RCM simulations only focused on the EASM. If this is not the case then further expand on the Indian monsoon. The regions selected for the RCM simulations have domain cut-offs near high topography regions most likely resulting in erroneous values. Also, it seems like adding china west to China1SE (Figure 4 analysis) would improve the RCM representation of the Indian Monsoon. The same problem occurs in section 3.4 the authors switch focus on the Indian monsoon but don't provide any nudged simulations of the EASM.

The authors used multiple model configurations with varying model resolutions and configurations. It seems important for the authors to note that increasing model resolutions can impact the regional circulation. This is particularly important when looking at a region that is strongly influenced by the regional topography. Add a section talking about the improvements and errors when increasing model resolution.

Lastly, the results incorporate many discussion points. For clarification, either add a separate discussion section or change the section to Results and Discussion as the title for section 3.

Technical corrections

Line 12 seamless modeling approach is vague. Perhaps adding a table of all the models, reanalysis, and observation used could help the readers.

Please be clear when using an abbreviation in the text. NWP or CPLDNWP should be stated in the text as coupled (CPLD) or uncoupled (UNCPLD) Northwest Pacific (NWP).

Consider adding a regular climatology figure for either GC2 or obs with some added labels/information of what the readers should focus on e.g. Indian and East Asian monsoon regions.

It seems like errors in GC2 are remedied in GloSea5. Perhaps the color bar needs to be adjusted since it would suggest that there are biases everywhere. The authors can also add a pattern correlation to clarify.

Paragraph 166 Is red warm SST bias and blue is cold bias? The text says Cold errors in the Arabian sea when I see red across the Somalia Jet region. Again a climatological figure or a description would help the readers. The same notation is used in the following paragraph.

Line 201 winter errors need a citation?

Line 215 consider the impact of changing model resolution over mountainous regions citations such as Curio et al., 2015, Acosta and Huber 2017, Anand et al., 2018

Figure 4 top panels are units the same as figure 1?

Figures 6 and 8 expand the region westward similarly to figure 7. Enhancement of warm SST anomalies over Somalia is cooccurring with the westward expansion of the EEIO cold anomalies should be noted.

Figure 9 add a caption for the red dashed box. Line 348 is the red dashed box in figure 9 northern EEIO?

Figure 13 and Line 350, should the readers focus on FOAM or OISSTV2 as the better model? Please state in the text why one would use FOAM. Why not show HadISST

like the rest of the analysis.

The comparison between CPLD and UNCPLD is interesting, the disparity in radiative fluxes during pre-monsoon should be further teased out.

Line 385 add a small explanation to identify the purpose of the selected regions.

Line 448 is vague and fully lets the ocean model off the hook. It should be further elaborated on and point out that an imbalance in net radiation fluxes leads to weak surface wind errors and is exacerbated by the inaccurate representation of the ocean mixed layer. Several studies have extensively studied the role of ocean heat transport and the authors should also note the role of land-ocean interaction which is not touched upon by the current study. See Chen and Bordoni 2014, and Park et al 2015 for EASM, and Lutsko etal 2019 for the Indian monsoon.

Line 452 again oceanic regions will not benefit from increased horizontal resolution however, many sections of the ASM region are over topography which will improve as you change model resolution.

Line 464 it should be noted that several similar works on the CMIP models have been done. See Sabeerali et al 2014, Anand et al 2018, Prasanna etal 2020, and Pathak et al 2019.

---

## Author Comment (AC1) · 15 Dec 2020

**GMD-2020-286**

Author response to comments by Anonymous Referee 1.

*Overall comments*

Using the MetUM unified modelling framework to decompose systematic modelling errors for the Asian Summer Monsoon is a wonderful example of the great utility of applying one framework to a science problem, in this case, quantifying systematic modelling errors in a monsoon system. The paper contains an enormous amount of information

that will be useful to modellers to improve skill for prediction (and projections) for the EASM. Although I appreciate the challenge of presenting this work in a concise and digestible way, I feel some improvements can be made, primarily to figure organization and better descriptions in the manuscript. To help guide the reader and improve the readability, I suggest the following: 1. To help guide the reader on experimental design, I recommend use of flowcharts for modelling tools and experiment description. 2. To help summarize the regional climate modelling results in Section 3.2, consider summary table (see specific comments). 3. To help digest and follow discussion on the initialized hindcasts in Section 3.3., consider reorganizing your figure suite such that specific locations (or errors foci in the text) are highlighted (see specific comments). If this is not possible, perhaps sub-heading per error topic, and better labelling on the figures will help.

We thank the reviewer for these helpful suggestions. We have added a Table of configurations to clarify each and how they related to one another, and a summary Table for the RCM results. We have divided the EASM and Indian Ocean analysis in section 3.3 under separate sub-headings and reordered the Figures accordingly. We have also separated the seasonal NWP hindcast analysis into separate sub-section 3.4. More details in answer to the specific comments below.

*Specific Comments*

Line 95: An explanation of N is needed for this grid system beyond what you have here. This will also help define the reader interpret "768" from N768 on Line 129, and N96 on Line 134.

In order to avoid confusion with our terminology, we have removed reference to N216 etc (except where used in a naming convention) and simply refer to the actual longitude x latitude grid resolution.

Figure 1: What is the reference vector?

A reference vector has been added.

Line 115: This isn't clear to me: Did you originally run RCM with GA6.0 physics? Or is your statement on a better Indian subcontinent simulation based on the GA6 vs GA7 comparison? I am not suggesting re-running anything, just a clarification on the justification for using GA7.0 configuration rather than that what was used for the global simulations. Also, to be clear, you used GA7.0 to force RCM (and not simply using the same model configuration)?

We apologise for the confusion caused by poor wording of this paragraph, which has been rewritten. The RCM was only configured with GA7.0, which differs from GA6.0 as mentioned here, but in which the overall pattern of ASM errors is very similar. The RCM is forced at the boundaries by 6-hourly ERA interim re-analysis. An additional corresponding 20-year atmosphere-only GCM simulation was run for comparison with this RCM configuration.

In response to a comment made by the other reviewer, we have added a Table showing the different model configurations used in this study.

Line 169 and Figure 2: The mean JJA cold bias of GloSea5 for parts of the Indian Ocean, around Malaysia, and perhaps Western Pacific look larger than the individual months?

This is because the JJA seasonal mean is from hindcasts initialised in April, so the lead time is longer for this plot. This has been clarified in the text.

Figure 3: The caption should note where domains overlap and also describe what N1 represents.

NI (no India) was not used in these experiments so this domain has been removed from Fig 3.
We have added information on the different domains, and how they overlap, to the caption. In addition, we have included the coordinates of the domains in the form $(x_0, y_0)(N_x, N_y)$ where $(x_0, y_0)$ is the position of the lower left hand corner of the region (in rotated pole coordinates) and $(N_x, N_y)$ is the number of grid points in the $x$ and $y$ direction.

Line 222 and Figure 4: It is hard to compare the GA7 GCM with Figure 1 top left with different color contours, scales, and vector arrows. It might be helpful to add a panel in this figure to truly compare the two. Also, it might be a good opportunity to discuss the improvements moving from GA6 to GA7 which would be interesting to readers of GMD.

We are reluctant to add yet another panel to this Figure; in addition, the GA7 GCM is atmosphere-only while Figure 1 top left is a coupled simulation. However, we have now reconciled the colour scales between Figures 4 and 5 and Figure 1, making the comparison easier. We have also calculated the pattern correlation between the rainfall errors in AGCM-N216 and those in GC2.0 for JJA (over the region shown in Fig 1 top left), which is 0.70. The changes between GA6 and GA7 are detailed in Walters et al. (2019, their section 4.2) so we do not go into detail here, but we have added an additional reference to this paper, and to an equivalent paper by Williams et al. (2017) for GC3 vs GC2, at the end of this paragraph.

Line 239: The westward extension in ChinaW seems to have a rather large impact over the Indian subcontinent. Explanation?

Extending the domain in China1W to include the Arabian Sea and part of the western equatorial Indian Ocean allows the dry bias over India and anti-cyclonic circulation bias to develop as it does in the GCM, while the circulation over the Indian subcontinent is very much constrained by reanalysis in China1. We have commented on this in the revised text.

Section 3.2: To elucidate the local/remote implications of each domain, one suggestion

would be to make a summary table, i.e. something like, one row per domain; one column for remote influence notations; one column for local influence notations.

We have included such a summary table as Table 2. In addition, we have re-drawn Figures 4 and 5 in such a way as to highlight the influence of the different domain extensions on the errors developing within the core China1 domain, by including differences from observations in the peripheral regions of the extended domains around the central domain (in which differences are shown against China1).

Figure 6: What is the reference vector?

A reference vector has now been added to the Figure.

Paragraph 259 and Figure 6: Comments on the dry biases in the Bay of Bengal?

See reply to next point.

Figure 7: What are your thoughts on what is going on in the Bay of Bengal. This cannot be explained by SSTs.

Thank you for noting that we should comment on this. It is related to the anticyclonic error over India which develops rapidly after initialisation and is associated with a weakening of the monsoon trough, combined with excessive rainfall over the steep orography of the eastern Himalaya that promotes convergence from the south and drying over the head of the Bay. Levine and Martin (2018) showed that the MetUM typically underestimates the number, and rainfall contribution from, monsoon lows and depressions, which also are unable to progress across northern India. In the absence of these features, rainfall over the Bay of Bengal is reduced and that over the Myanmar orography is increased, with an associated acceleration of the westerly flow across the Bay of Bengal and SE Asia into the South China Sea. This converges with the southerly anomalies from the Maritime Continent region, promoting further rainfall and creating a positive feedback that develops a westerly wind error (extension of the westerly jet) across the SCS and the Philippines into the western Pacific.

In the head of the Bay, we think the SSTs in the coupled model warm in response to the reduced rainfall and cloud and to convergence of warm low-level winds from northern India, while further south, as we show subsequently, the SSTs respond to these changes by (ultimately) cooling. Both are likely to be exacerbated by an ocean mixed layer that is too shallow. It appears to be mostly the atmosphere that is driving the ocean here, with limited compensating feedback, although further sensitivity tests will be needed to confirm this, and these will be the subject of future work.

We have added comments on these features to the text in section 3.3 and in the Summary.

Line 280: The N/S dipole seems weak.

We have noted this in the text.

Section 3.3: It is hard to follow specific locations for much of the discussion. I recommend picking a few key areas and designing your figures (6-9) around specific locations/error sources (i.e. one location/error per figure but include the information contained across 6 – 9 but also 10-13) This might help to clearly show the progressions and biases. For example, the South China sea area, or the Bay of Bengal, or EEIO. Full plots as shown here can be supplemental for readers interested in something the authors do not highlight, but for the discussion explicitly called out in the text, there needs to be better organization of figures.

We appreciate the point made by the reviewer here, but we feel it is important to show how the regional-scale errors fit into the wider pattern, and we are also keen to avoid increasing the number of figures too much. For the Indian Ocean region, however, we agree that zooming in would be helpful. We have therefore kept the full plots as they were for Figures 6, 7 and 10 (and actually extended the panels in Fig.s 6 and 10 (now Fig. 9) westwards to match the region shown in Fig. 7 as requested by the other reviewer, but reduced the region plotted for Fig.s 8 and 9 (now Fig.s 10 and 11). We have also reorganised section 3.3 under different sub-headings in order to

focus the reader on each particular region, and separated out the analysis using the NWP hindcasts (which largely focusses on the EEIO as an example) into an additional sub-section 3.4.

Figures 7 and 9: Please define all components of the figures in the captions or note them in the text. I don't see an explanation of the red dashed box?

This has been corrected.

Line 306: Define SCSSM.

South China Sea Summer Monsoon – this has been expanded in the text.

Figure 11d: I feel like there is much to unpack from this panel beyond the few paragraphs in the text. I see that the dashed/dashed-dotted lines are defined in the caption, but some attention to these should be paid in the text with further explanation as to interpretation.

There was already some discussion on this in lines 312-315 of the original manuscript, but we agree that more detail is warranted. Additional discussion of the EASMI panel has been added in the new subsection 3.3.2.

Figure 12: Shading = color contours? What is the reference vector?

Yes shading refers to the colour scale. This, and a reference vector, have been added.

Lines 351,357,370 and Figure 13: CPLDNW, UNCPLD, and FOAM, although we can guess, should be explicitly defined.

This has been done, both in the text and through the addition of Table 1 which details the configurations used.

Figure 14: Labelling and boxes should be cleaned up and consistent.

We have removed Figure 14 as the boxes are shown on the subsequent Figure (formerly Fig. 15, now Fig. 14).

Figure 16: What is the reference vector?

Reference vector has been added.

---

## Author Comment (AC2) · 15 Dec 2020

**GMD-2020-268**

Response to comments by Anonymous Referee 2.

This article develops a systematic method to detect biases in model simulations to improve the representation of various features of the Asian summer monsoon system in climate models. The study used multiple configurations of the Met Office Unified Model which encompass global climate simulations (fully coupled and atmosphere only), regional climate simulations, and regional nudging simulations. The authors focused on ocean-centric regions such as the Indian Ocean, Maritime Continent, and

the Philippines to demonstrate the growth of regional erroneous atmospheric-ocean circulation over time that can impact the Asian summer monsoon system.

*General comments*

By providing a multi-layered framework when simulating the Asian monsoon system the authors identified key ocean regions that produce systematic errors and that if corrected will improve how climate and weather models simulate the system. That alone is crucial for the field and although this framework is overall beneficial, the presentation of the text and results could be further improved.

*Specific Comments*

The title and abstract could use some refocusing - no need to mention Asian monsoon if your main goal is only the EASM. Otherwise, the authors should add some minor additional work to fully represent the title. The authors should restructure the manuscript into EASM, SASM, and Southeast Asia analyses.

We thank the reviewer for this suggestion. However, the aim of this study is to demonstrate the tools and techniques that can be employed in a seamless modelling system to elucidate the source of systematic errors. The ASM is a large system that includes several regional, but interacting, monsoons. Therefore, in demonstrating these tools and techniques, we consider the ASM as a whole but use examples taken from some (not all) of the regional monsoons within this. We would argue that it is not necessary to divide the manuscript into separate analysis of the regional monsoons in order to fulfil the aims of our study. Doing so would both add unnecessarily to the length of the manuscript and potentially lead to confusion as to which parts of the system relate to which regional monsoon (often more than one).

**[GMDD](https://www.geosci-model-dev-discuss.net)**
[Figure]

We have, however, reorganised section 3.3 under different sub-headings and separated out the analysis using the NWP hindcasts (which largely focusses the EEIO as an example) into an additional sub-section 3.4. We hope that this will allow the reader to focus on each region within the context of the ASM as a whole.

The RCM simulations only focused on the EASM. If this is not the case then further expand on the Indian monsoon. The regions selected for the RCM simulations have domain cut-offs near high topography regions most likely resulting in erroneous values. Also, it seems like adding china west to China1SE (Figure 4 analysis) would improve the RCM representation of the Indian Monsoon. The same problem occurs in section3.4 the authors switch focus on the Indian monsoon but don't provide any nudged simulations of the EASM.

RCM simulations centred on the Indian Monsoon region were published previously in studies by Karmacharya et al. (2015) and Levine and Martin (2018), hence they are not included here as we wished to limit the number of examples given of the use of this technique. However, we have perhaps not made sufficient reference to these earlier studies nor their findings, particularly on how the inclusion of East Asia in the domain centred over India affects the SASM. Levine and Martin (2018) showed that such an extension made very little difference to the mean state errors over India. We have added more information on this to section 3.2.

The domain cut-offs near high topography regions do result in erroneous values, but this is difficult to avoid without either including or excluding the entire Himalayas and Tibetan Plateau from the domains, which would result in domains covering a huge area or domains only covering E China. The existing domain cut-offs mainly result in erroneous values locally in the boundary regions, and this is far enough away from the area of interest (China in this case).

Extending the RCM domain westwards (and southwards), as in China1W and China1SW, results in a poor simulation of the Indian Monsoon by including within the
RCM areas such as western India, the equatorial Indian Ocean and Arabian Sea that are responsible for a large part for the large Indian Monsoon biases in the GCM. Therefore it is unlikely that the extending China1SE westwards would improve either the Indian Monsoon simulation or the EASM (except perhaps through error compensation).

The simulations where nudging is applied to the Philippines and Indonesia regions directly relate to understanding the errors that affect the EASM. Nudging over the EASM region itself would only provide information about the influence of this region on the other parts of the ASM system. The inclusion of the simulation with nudging over the SASM region is in order to demonstrate that (in contrast to the EASM region) much of the error pattern in the SASM region develops locally and that it also influences the wider ASM system. We have now clarified the reasons behind the choice of nudging regions at the start of section 3.5.1.

The authors used multiple model configurations with varying model resolutions and configurations. It seems important for the authors to note that increasing model resolutions can impact the regional circulation. This is particularly important when looking at a region that is strongly influenced by the regional topography. Add a section talking about the improvements and errors when increasing model resolution.

There are several studies (e.g. Johnson et al., 2016) which demonstrate that the robust systematic errors in monsoon simulations are largely unaffected by model resolution, despite some small impacts on the regional detail. Analysis of the NWP hindcasts also shows similar the error evolution to that seen in GloSea5 despite their significant increase in horizontal resolution. We have separated the latter analysis into a separate sub-section 3.4.1, and we have added some addition detail on this at the end of the Summary.

Lastly, the results incorporate many discussion points. For clarification, either add a separate discussion section or change the section to Results and Discussion as the title for section 3.

We have altered the title for section 3 as suggested, and separated the summary and conclusions into separate sections with more detail in the former, in order to bring things together.

*Technical corrections*

Line 12 seamless modeling approach is vague. Perhaps adding a table of all the models, reanalysis, and observation used could help the readers.

A table of configurations has now been added (Table 1).

Please be clear when using an abbreviation in the text. NWP or CPLDNWP should be stated in the text as coupled (CPLD) or uncoupled (UNCPLD) Northwest Pacific (NWP).

NWP is Numerical Weather Prediction, as defined in section 2. We have made changes throughout the revised manuscript to clarify the abbreviations.

Consider adding a regular climatology figure for either GC2 or obs with some added labels/information of what the readers should focus on e.g. Indian and East Asian monsoon regions.

The ASM region is well-known to modellers, and we would prefer not to increase the already-large number of figures in our manuscript. Instead, we have referenced key publications relating to ASM errors in the introduction, to help locate the readers.

It seems like errors in GC2 are remedied in GloSea5. Perhaps the color bar needs to be adjusted since it would suggest that there are biases everywhere. The authors can also add a pattern correlation to clarify.

We feel it is essential to use the same colour bar for both, in order to demonstrate that the error patterns are not that different, although the magnitudes are slightly reduced

in GloSea5, as we mentioned in the manuscript. We have used very faint colours for the values between -2.5 – 2.5 mm/day. These could be replaced with no colour, but we prefer to retain the slight distinction between positive and negative values. Pattern correlations have been added to the text. With the exception of June (where the errors in GC2 are larger and somewhat altered, due to the development of SST errors over longer timescales, as discussed) these are above 0.8 for rainfall, and between 0.6 and 0.7 for SST.

Paragraph 166 Is red warm SST bias and blue is cold bias? The text says Cold errors in the Arabian sea when I see red across the Somalia Jet region. Again a climatological figure or a description would help the readers. The same notation is used in the following paragraph.

The cold bias (blue) is evident in the northern Arabian Sea, whereas the warm bias (red) further south is part of the broader western Indian Ocean warm bias. We have clarified that we are referring to the northern Arabian Sea in the text.

Line 201 winter errors need a citation?

This refers to the discussion in previous paragraphs, in which citations have already been provided. We have noted that in this line of the revised manuscript.

Line 215 consider the impact of changing model resolution over mountainous regions, citations such as Curio et al., 2015, Acosta and Huber 2017, Anand et al., 2018

As noted above, several previous studies have demonstrated that the systematic errors in monsoon simulations using this model are largely unaffected by model resolution, despite some small impacts on the regional detail. We now mention this explicitly in new sub-section 3.4.1.

Figure 4 top panels are units the same as figure 1?

Figures 4 and 5 have been redrawn with a colour scale and wind vector that matches Figure 1.

Figures 6 and 8 expand the region westward similarly to figure 7. Enhancement of warm SST anomalies over Somalia is cooccurring with the westward expansion of the EEIO cold anomalies should be noted.

We have expanded this region to the west in Figures 6 and 8 (now Fig. 10), but also reduced the eastward extent of Figures 8 and 9 (now figures 10 and 11) in order to focus on the Indian Ocean.

The development of warm SST anomalies in the western Indian Ocean as the cold anomalies develop and expand from the east was mentioned in the initial discussion of Figure 7 but is now reiterated in subsection 3.3.3 which discusses the equatorial Indian Ocean specifically.

Figure 9 add a caption for the red dashed box. Line 348 is the red dashed box in figure 9 northern EEIO?

This has been clarified.

Figure 13 and Line 350, should the readers focus on FOAM or OISSTV2 as the better model? Please state in the text why one would use FOAM. Why not show HadISST like the rest of the analysis.

We include FOAM SSTs in Fig. 13 because these analyses are used to initialise the CPLDNWP hindcasts. They differ from the OISSTv2 SSTs because they are analyses rather than observations. All of the other Figures except Fig. 2 used OISSTv2 (Reynolds) so we have now replaced Figure 2 with differences against OISSTv2 for consistency.

The comparison between CPLD and UNCPLD is interesting, the disparity in radiative fluxes during pre-monsoon should be further teased out.

We agree, but this is beyond the scope of the current study, which seeks to demonstrate the use of these techniques rather than to understand fully all of the details. We have already noted in the text that this is partly related to the near-surface wind error and

excessive surface latent heat flux, and that further work is required to understand these coupled feedbacks. This may involve the use of targeted sensitivity tests in order to separate the different components. We have added the latter point to the revised text.

Line 385 add a small explanation to identify the purpose of the selected regions.

We have added an explanation to the text at the start of section 3.5.1.

Line 448 is vague and fully lets the ocean model off the hook. It should be further elaborated on and point out that an imbalance in net radiation fluxes leads to weak surface wind errors and is exacerbated by the inaccurate representation of the ocean mixed layer. Several studies have extensively studied the role of ocean heat transport and the authors should also note the role of land-ocean interaction which is not touched upon by the current study. See Chen and Bordoni 2014, and Park et al 2015 for EASM, and Lutsko et al 2019 for the Indian monsoon.

We agree that we have not worded this very fairly, and we thank the reviewer for the additional references. We have reworded and expanded this point in the revised text.

Line 452 again oceanic regions will not benefit from increased horizontal resolution however, many sections of the ASM region are over topography which will improve as you change model resolution.

There is minimal evidence that increasing the model resolution improves the ASM region over topography substantially. Johnson et al (2016) showed that, while there are a number of small, beneficial impacts from increasing resolution in the MetUM, it does not solve the many monsoon biases. We have added a reference to this work in this line, and also noted more explicitly in section 3.4.1 the evidence for this statement from the current study.

Line 464 it should be noted that several similar works on the CMIP models have been done. See Sabeerali et al 2014, Anand et al 2018, Prasanna et al 2020, and Pathak et al 2019.

While there have been several works that examine systematic errors in the Asian summer monsoon in CMIP models (including those that you reference here), and some which use initialized modelling frameworks to diagnose the origins of such errors (such as those referenced in the Introduction), the use of a range of techniques such as those described here within a seamless modelling system that includes both coupled and atmosphere-only configurations and regional modelling to analyse the development and sources of particular errors on a range of timescales has not, to our knowledge, been demonstrated.

---

## Author Comment (AC3) · 15 Dec 2020

**Understanding the development of systematic errors in the Asian Summer Monsoon**

Gill M. Martin[1], Richard C. Levine[1], José M. Rodriguez[1] and Michael Vellinga[1]

[1]Met Office, Exeter, Devon, EX1 3PB, UK

5 *Correspondence to*: Gill M Martin (gill.martin@metoffice.gov.uk)

**Abstract.** Despite the importance of monsoon rainfall to over half of the world's population, many of the current generation of climate models struggle to capture some of the major features of the various monsoon systems. Studies of the development of errors in several tropical regions have shown that they start to develop very quickly, within the first few days of a model simulation, and can then persist to climate timescales. Understanding the sources of such errors requires the

10 combination of various modelling techniques and sensitivity experiments of varying complexity. Here, we demonstrate how such analysis can shed light on the way in which monsoon errors develop, their local and remote drivers and feedbacks. We make use of the seamless modelling approach adopted by the Met Office, whereby different applications of the Met Office Unified Model (MetUM) use essentially the same model configuration (dynamical core and physical parametrisations) across a range of spatial and temporal scales. Using the Asian Summer Monsoon (ASM) as an example, we show that error

15 patterns in circulation and rainfall over the ASM region in the MetUM are similar between multi-decadal climate simulations and seasonal hindcasts initialised in spring. Analysis of the development of these errors on both short-range and seasonal timescales following model initialisation suggests that both the Maritime Continent and the oceans around the Philippines play a role in the development of  East Asia summer monsoon errors, with the Indian summer monsoon region providing an additional contribution, while the errors over the Indian summer monsoon

20 region itself appear to arise locally. Regional modelling with various lateral boundary locations helps to separate local and remote contributions to the errors, while regional relaxation experiments shed light on the influence of errors developing within particular areas on the region as a whole.

**Copyright statement**

25 The works published in this journal are distributed under the Creative Commons Attribution 4.0 License. This licence does not affect the Crown copyright work, which is re-usable under the Open Government Licence (OGL). The Creative Commons Attribution 4.0 License and the OGL are interoperable and do not conflict with, reduce or limit each other.

© Crown copyright 2020

[revised manuscript text omitted]

---

## Author Response (AR2)

**Response to Topical Editor's report on "Understanding the development of systematic errors in the Asian Summer Monsoon" by Gill M. Martin et al.**

Topical Editor Decision: Publish subject to minor revisions (review by editor) (23 Dec 2020) by Richard Neale

Comments to the Author:

Hi Gill,

Hope you are well.

So it seems the authors are mostly happy to accept with minor revisions which seems reasonable, so congrats.

It seems there were some concerns regarding the presentation clarity and the organization/description of the experiments. Recommendations for including a more visual flow chart of the experiments and some discussions of the role of resolution were made, so I would encourage you to address these as best as possible.

Best regards

Rich Neale

We thank the Topical Editor for these positive comments. In our first revision we attempted to address the concerns expressed by the reviewers regarding the clarity and organization of the manuscript. This included adding a table summarising the different model configurations, dividing the EASM and Indian Ocean analysis in section 3.3 under separate sub-headings and reordering the figures accordingly, separating the seasonal NWP hindcast analysis into a separate sub-section, and adding a separate sub-section on the role of horizontal resolution. We hope that this was sufficient to improve the clarity but please let us know if further specific changes are required.

We apologise for not previously including a flow chart of the methodology. We have now included this as Figure 16 and referred to it at the start of the Summary section. We hope that this helps to illustrate our approach better.

Best wishes,

Gill